# Flow Network based Generative Models for Non-Iterative Diverse Candidate Generation

**Emmanuel Bengio**[1,2]**, Moksh Jain**[1,5]**, Maksym Korablyov**[1]
**Doina Precup**[1,2,4]**, Yoshua Bengio**[1,3]
[1]Mila, [2]McGill University, [3]Université de Montréal, [4]DeepMind, [5]Microsoft

## Abstract

This paper is about the problem of learning a stochastic policy for generating an object (like a molecular graph) from a sequence of actions, such that the probability of generating an object is proportional to a given positive reward for that object. Whereas standard return maximization tends to converge to a single return-maximizing sequence, there are cases where we would like to sample a diverse set of high-return solutions. These arise, for example, in black-box function optimization when few rounds are possible, each with large batches of queries, where the batches should be diverse, e.g., in the design of new molecules. One can also see this as a problem of approximately converting an energy function to a generative distribution. While MCMC methods can achieve that, they are expensive and generally only perform local exploration. Instead, training a generative policy amortizes the cost of search during training and yields to fast generation. Using insights from Temporal Difference learning, we propose GFlowNet, based on a view of the generative process as a flow network, making it possible to handle the tricky case where different trajectories can yield the same final state, e.g., there are many ways to sequentially add atoms to generate some molecular graph. We cast the set of trajectories as a flow and convert the flow consistency equations into a learning objective, akin to the casting of the Bellman equations into Temporal Difference methods. We prove that any global minimum of the proposed objectives yields a policy which samples from the desired distribution, and demonstrate the improved performance and diversity of GFlowNet on a simple domain where there are many modes to the reward function, and on a molecule synthesis task.

## 1 Introduction

The maximization of expected return $R$ in reinforcement learning (RL) is generally achieved by putting all the probability mass of the policy $\pi$ on the highest-return sequence of actions. In this paper, we study the scenario where our objective is not to generate the single highest-reward sequence of actions but rather to sample a distribution of trajectories whose probability is proportional to a given positive return or reward function. This can be useful in tasks where exploration is important, i.e., we want to sample from the leading modes of the return function. This is equivalent to the problem of turning an energy function into a corresponding generative model, where the object to be generated is obtained via a sequence of actions. By changing the temperature of the energy function (i.e., scaling it multiplicatively) or by taking the power of the return, one can control how selective the generator should be, i.e., only generate from around the highest modes at low temperature or explore more with a higher temperature.

A motivating application for this setup is iterative black-box optimization where the learner has access to an oracle which can compute a reward for a large batch of candidates at each round, e.g., in drug-discovery applications. Diversity of the generated candidates is particularly important when the oracle is itself uncertain, e.g., it may consist of cellular assays which is a cheap proxy for clinical

35th Conference on Neural Information Processing Systems (NeurIPS 2021).

trials, or it may consist of the result of a docking simulation (estimating how well a candidate small molecule binds to a target protein) which is a proxy for more accurate but more expensive downstream evaluations (like cellular assays or in-vivo assays in mice).

When calling the oracle is expensive (e.g. it involves a biological experiment), a standard way (Anger-mueller et al., 2020) to apply machine learning in such exploration settings is to take the data already collected from the oracle (say a set of $(x, y)$ pairs where $x$ is a candidate solution an $y$ is a scalar evaluation of $x$ from the oracle) and train a supervised proxy $f$ (viewed as a simulator) which predicts $y$ from $x$. The function $f$ or a variant of $f$ which incorporates uncertainty about its value, like in Bayesian optimization (Srinivas et al., 2010; Negoescu et al., 2011), can then be used as a reward function $R$ to train a generative model or a policy that will produce a batch of candidates for the next experimental assays. Searching for $x$ which maximizes $R(x)$ is not sufficient because we would like to sample for the batch of queries a representative set of $x$'s with high values of $R$, i.e., around modes of $R(x)$. Note that alternative ways to obtain diversity exist, e.g., with batch Bayesian optimization (Kirsch et al., 2019). An advantage of the proposed approach is that the computational cost is linear in the size of the batch (by opposition with methods which compare pairs of candidates, which is at least quadratic). With the possibility of assays of a hundred thousand candidates using synthetic biology, linear scaling would be a great advantage.

**In this paper, we thus focus on the specific machine learning problem of turning a given positive reward or return function into a generative policy which samples with a probability proportional to the return.** In applications like the one mentioned above, we only apply the reward function after having generated a candidate, i.e., the reward is zero except in a terminal state, and the return is the terminal reward. We are in the so-called episodic setting of RL.

The proposed approach views the probability assigned to an action given a state as the flow associated with a network whose nodes are states, and outgoing edges from that node are deterministic transitions driven by an action (not to be confused with normalizing flows; Rezende and Mohamed (2016)). The total flow into the network is the sum of the rewards in the terminal states (i.e., a partition function) and can be shown to be the flow at the root node (or start state). The proposed algorithm is inspired by Bellman updates and converges when the incoming and outgoing flow into and out of each state match. A policy which chooses an action with probability proportional to the outgoing flow corresponding to that action is proven to achieve the desired result, i.e., the probability of sampling a terminal state is proportional to its reward. In addition, we show that the resulting setup is off-policy; it converges to the above solution even if the training trajectories come from a different policy, so long as it has large enough support on the state space.

The main contributions of this paper are as follows:

- We propose GFlowNet, a novel generative method for unnormalized probability distributions based on flow networks and local flow-matching conditions: the flow incoming to a state must match the outgoing flow.

- We prove crucial properties of GFlowNet, including the link between the flow-matching conditions (which many training objectives can provide) and the resulting match of the generated policy with the target reward function. We also prove its offline properties and asymptotic convergence (if the training objective can be minimized). We also demonstrate that previous related work (Buesing et al., 2019) which sees the generative process like a tree would fail when there are many action sequences which can lead to the same state.

- We demonstrate on synthetic data the usefulness of departing from seeking one mode of the return, and instead seeking to model the entire distribution and all its modes.

- We successfully apply GFlowNet to a large scale molecule synthesis domain, with comparative experiments against PPO and MCMC methods.

All implementations are available at https://github.com/bengioe/gflownet.

## 2   Approximating Flow Network generative models with a TD-like objective

Consider a discrete set $\mathcal{X}$ and policy $\pi(a|s)$ to sequentially build $x \in \mathcal{X}$ with probability $\pi(x)$ with

$$\pi(x) \approx \frac{R(x)}{Z} = \frac{R(x)}{\sum_{x' \in \mathcal{X}} R(x')} \tag{1}$$

where $R(x) > 0$ is a reward for a terminal state $x$. This would be useful to sample novel drug-like molecules when given a reward function $R$ that scores molecules based on their chemical properties. Being able to sample from the high modes of $R(x)$ would provide diversity in the batches of generated molecules sent to assays. This is in contrast with the typical RL objective of maximizing return which we have found to often end up focusing around one or very few good molecules. In our context, $R(x)$ is a proxy for the actual values obtained from assays, which means it can be called often and cheaply. $R(x)$ is retrained or fine-tuned each time we acquire new data from the assays.

What method should one use to generate batches sampled from $\pi(x) \propto R(x)$? Let's first think of the state space under which we would operate.

Let $\mathcal{S}$ denote the set of states and $\mathcal{X} \subset \mathcal{S}$ denote the set of terminal states. Let $\mathcal{A}$ be a finite set, the alphabet, $\mathcal{A}(s) \subseteq \mathcal{A}$ be the set of allowed actions at state $s$, and let $\mathcal{A}^*(s)$ be the set of all sequences of actions allowed after state $s$. To every action sequence $\vec{a} = (a_1, a_2, a_3, ..., a_h)$ of $a_i \in \mathcal{A}, h \leq H$ corresponds a single $x$, i.e. the environment is deterministic so we can define a function $C$ mapping a sequence of actions $\vec{a}$ to an $x$. If such a sequence is 'incomplete' we define its reward to be 0. When the correspondence between action sequences and states is **bijective**, a state $s$ is uniquely described by some sequence $\vec{a}$, and we can visualize the generative process as the traversal of a tree from a single root node to a leaf corresponding to the sequence of actions along the way.

However, when this correspondence is **non-injective**, i.e. when multiple action sequences describe the same $x$, things get trickier. Instead of a tree, we get a directed acyclic graph or DAG (assuming that the sequences must be of finite length, i.e., there are no deterministic cycles), as illustrated in Figure 1. For example, and of interest here, molecules can be seen as graphs, which can be described in multiple orders (canonical representations such as SMILES strings also have this problem: there may be multiple descriptions for the same actual molecule). The standard approach to such a sampling problem is to use iterative MCMC methods (Xie et al., 2021; Grathwohl et al., 2021). Another option is to relax the desire to have $p(x) \propto R(x)$ and to use non-interative (sequential) RL methods (Gottipati et al., 2020), but these are at high risk of getting stuck in local maxima and of missing modes. Indeed, in our setting, the policy which maximizes the expected return (which is the expected final reward) generates the sequence with the highest return (i.e., a single molecule).

## 2.1 Flow Networks

In this section we propose the Generative Flow Network framework, or GFlowNet, which enables us to learn policies such that $p(x) \propto R(x)$ when sampled. We first discuss why existing methods are inadequate, and then show how we can use the metaphor of flows, sinks and sources, to construct adequate policies. We then show that such policies can be learned via a flow-matching objective.

With existing methods in the bijective case, one can think of the sequential generation of one $x$ as an episode in a tree-structured deterministic MDP, where all leaves $x$ are terminal states (with reward $R(x)$) and the root is initial state $s_0$. Interestingly, in such a case one can express the pseudo-value of a state $\tilde{V}(s)$ as the sum of all the rewards of the descendants of $s$ (Buesing et al., 2019).

In the non-injective case, these methods are inadequate. Constructing $\pi(\tau) \approx R(\tau)/Z$, e.g. as per Buesing et al. (2019), MaxEnt RL (Haarnoja et al., 2017), or via an autoregressive method (Nash and Durkan, 2019; Shi et al., 2021) has a particular problem as shown below: if multiple action sequences $\vec{a}$ (i.e. multiple trajectories $\tau$) lead to a final state $x$, then a serious bias can be introduced in the generative probabilities. Let us denote $\vec{a} + \vec{b}$ as the concatenation of the two sequences of actions $\vec{a}$ and $\vec{b}$, and by extension $s + \vec{b}$ the state reached by applying the actions in $\vec{b}$ from state $s$.

**Proposition 1.** *Let $C : \mathcal{A}^* \mapsto \mathcal{S}$ associate each allowed action sequence $\vec{a} \in \mathcal{A}^*$ to a state $s = C(\vec{a}) \in \mathcal{S}$. Let $\tilde{V} : \mathcal{S} \mapsto \mathbf{R}^+$ associate each state $s \in \mathcal{S}$ to $\tilde{V}(s) = \sum_{\vec{b} \in \mathcal{A}^*(s)} R(s + \vec{b}) > 0$, where $\mathcal{A}^*(s)$ is the set of allowed continuations from $s$ and $s + \vec{b}$ denotes the resulting state, i.e., $\tilde{V}(s)$ is the sum of the rewards of all the states reachable from $s$. Consider a policy $\pi$ which starts from the state corresponding to the empty string $s_0 = C(\emptyset)$ and chooses from state $s \in \mathcal{S}$ an allowable action $a \in \mathcal{A}(s)$ with probability $\pi(a|s) = \frac{\tilde{V}(s+a)}{\sum_{b \in \mathcal{A}(s)} \tilde{V}(s+b)}$. Denote $\pi(\vec{a} = (a_1, \ldots, a_N)) = \prod_{i=1}^N \pi(a_i | C(a_1, \ldots, a_{i-1}))$ and $\pi(s)$ with $s \in \mathcal{S}$ the probability of visiting a state $s$ with this policy. The following then obtains:*
*(a) $\pi(s) = \sum_{\vec{a}_i : C(\vec{a}_i) = s} \pi(\vec{a}_i)$.*

*(b) If $C$ is bijective, then $\pi(s) = \frac{\tilde{V}(s)}{\tilde{V}(s_0)}$ and as a special case for terminal states $x$, $\pi(x) = \frac{R(x)}{\sum_{x \in \mathcal{X}} R(x)}$.*
*(c) If $C$ is non-injective and there are $n(x)$ distinct action sequences $\vec{a}_i$ s.t. $C(\vec{a}_i) = x$, then $\pi(x) = \frac{n(x)R(x)}{\sum_{x' \in \mathcal{X}} n(x')R(x')}$.*

See Appendix A.1 for the proof. In combinatorial spaces, such as for molecules, where $C$ is non-injective (there are many ways to construct a molecule graph), this can become exponentially bad as trajectory lengths increase. It means that larger molecules would be exponentially more likely to be sampled than smaller ones, just because of the many more paths leading to them. In this scenario, the pseudo-value $\tilde{V}$ is "misinterpreting" the MDP's structure as a tree, leading to the wrong $\pi(x)$.

An alternative is to see the MDP as a **flow network**, that is, leverage the DAG structure of the MDP, and learn a flow $F$, rather than estimating the pseudo-value $\tilde{V}$ as a sum of descendant rewards, as elaborated below. We define the flow network as a having a single source, the root node (or initial state) $s_0$ with in-flow $Z$, and one sink for each leaf (or terminal state) $x$ with out-flow $R(x) > 0$. We write $T(s, a) = s'$ to denote that the state-action pair $(s, a)$ leads to state $s'$. Note that because $C$ is not a bijection, i.e., there are many paths (action sequences) leading to some node, a node can have multiple parents, i.e. $|\{(s, a) \mid T(s, a) = s'\}| \geq 1$, except for the root, which has no parent. We write $F(s, a)$ for the flow between node $s$ and node $s' = T(s, a)$, $F(s)$ for the total flow going through $s$.[1] This construction is illustrated in Fig. 1.

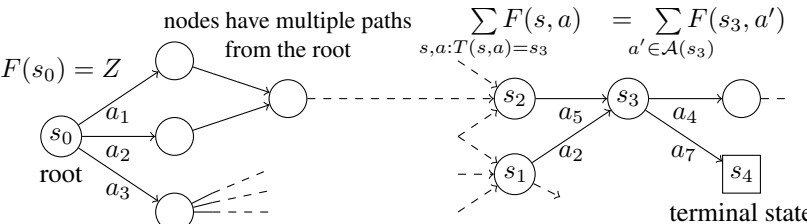

Figure 1: A flow network MDP. Episodes start at source $s_0$ with flow $Z$. Like with SMILES strings, there are no cycles. Terminal states are sinks with out-flow $R(s)$. Exemplar state $s_3$ has parents $\{(s, a) | T(s, a) = s_3\} = \{(s_1, a_2), (s_2, a_5)\}$ and allowed actions $\mathcal{A}(s_3) = \{a_4, a_7\}$. $s_4$ is a terminal sink state with $R(s_4) > 0$ and only one parent. The goal is to estimate $F(s, a)$ such that the flow equations are satisfied for all states: for each node, incoming flow equals outgoing flow.

To satisfy flow conditions, we require that for any node, the incoming flow equals the outgoing flow, which is the total flow $F(s)$ of node $s$. Boundary conditions are given by the flow into the terminal nodes $x$, $R(x)$. Formally, for any node $s'$, we must have that the in-flow

$$F(s') = \sum_{s,a:T(s,a)=s'} F(s, a) \tag{2}$$

equals the out-flow

$$F(s') = \sum_{a' \in \mathcal{A}(s')} F(s', a'). \tag{3}$$

More concisely, with $R(s) = 0$ for interior nodes, and $\mathcal{A}(s) = \varnothing$ for leaf (sink/terminal) nodes, we write the following *flow consistency equations*:

$$\sum_{s,a:T(s,a)=s'} F(s, a) = R(s') + \sum_{a' \in \mathcal{A}(s')} F(s', a'). \tag{4}$$

with $F$ being a flow, $F(s, a) > 0 \ \forall s, a$ (for this we needed to constrain $R(x)$ to be positive too). One could include in principle nodes and edges with zero flow but it would make it difficult to talk about the logarithm of the flow, as we do below, and such states can always be excluded by the allowed set of actions for their parent states. Let us now show that such a flow correctly produces $\pi(x) = R(x)/Z$ when the above flow equations are satisfied.

---

[1] In some sense, $F(s)$ and $F(s, a)$ are close to $V$ and $Q$, RL's value and action-value functions. These effectively inform an agent taking decisions at each step of an MDP to act in a desired way. With some work, we can also show an equivalence between $F(s, a)$ and the "real" $Q^{\hat{\pi}}$ of some policy $\hat{\pi}$ in a modified MDP (see A.2).

**Proposition 2.** *Let us define a policy $\pi$ that generates trajectories starting in state $s_0$ by sampling actions $a \in \mathcal{A}(s)$ according to*

$$\pi(a|s) = \frac{F(s,a)}{F(s)} \tag{5}$$

*where $F(s,a) > 0$ is the flow through allowed edge $(s,a)$, $F(s) = R(s) + \sum_{a \in \mathcal{A}(s)} F(s,a)$ where $R(s) = 0$ for non-terminal nodes $s$ and $F(x) = R(x) > 0$ for terminal nodes $x$, and the flow consistency equation $\sum_{s,a:T(s,a)=s'} F(s,a) = R(s') + \sum_{a' \in \mathcal{A}(s')} F(s',a')$ is satisfied. Let $\pi(s)$ denote the probability of visiting state $s$ when starting at $s_0$ and following $\pi(\cdot|\cdot)$. Then*
*(a) $\pi(s) = \frac{F(s)}{F(s_0)}$*
*(b) $F(s_0) = \sum_{x \in \mathcal{X}} R(x)$*
*(c) $\pi(x) = \frac{R(x)}{\sum_{x' \in \mathcal{X}} R(x')}$.*

*Proof.* We have $\pi(s_0) = 1$ since we always start in root node $s_0$. Note that $\sum_{x \in \mathcal{X}} \pi(x) = 1$ because terminal states are mutually exclusive, but in the case of non-bijective $C$, we cannot say that $\sum_{s \in \mathcal{S}} \pi(s)$ equals 1 because the different states are not mutually exclusive in general. This notation is different from the one typically used in RL where $\pi(s)$ refers to the asymptotic distribution of the Markov chain. Then

$$\pi(s') = \sum_{(a,s):T(s,a)=s'} \pi(a|s)\pi(s) \tag{6}$$

i.e., using Eq. 5,

$$\pi(s') = \sum_{(a,s):T(s,a)=s'} \frac{F(s,a)}{F(s)}\pi(s). \tag{7}$$

We can now conjecture that the statement

$$\pi(s) = \frac{F(s)}{F(s_0)} \tag{8}$$

is true and prove it by induction. This is trivially true for the root, which is our base statement, since $\pi(s_0) = 1$. By induction, we then have that if the statement is true for parents $s$ of $s'$, then

$$\pi(s') = \sum_{s,a:T(s,a)=s'} \frac{F(s,a)}{F(s)} \frac{F(s)}{F(s_0)} = \frac{\sum_{s,a:T(s,a)=s'} F(s,a)}{F(s_0)} = \frac{F(s')}{F(s_0)} \tag{9}$$

which proves the statement, i.e., the first conclusion (a) of the theorem. We can then apply it to the case of terminal states $x$, whose flow is fixed to $F(x) = R(x)$ and obtain

$$\pi(x) = \frac{R(x)}{F(s_0)}. \tag{10}$$

Noting that $\sum_{x \in X} \pi(x) = 1$ and summing both sides of Eq. 10 over $x$ we thus obtain (b), i.e., $F(s_0) = \sum_{x \in \mathcal{X}} R(x)$. Plugging this back into Eq. 10, we obtain (c), i.e., $\pi(x) = \frac{R(x)}{\sum_{x' \in \mathcal{X}} R(x')}$. $\square$

Thus our choice of $\pi(a|s)$ satisfies our desiderata: it maps a reward function $R$ to a generative model which generates $x$ with probability $\pi(x) \propto R(x)$, whether $C$ is bijective or non-injective (the former being a special case of the latter, and we just provided a proof for the general non-injective case).

## 2.2 Objective Functions for GFlowNet

We can now leverage our RL intuitions to create a learning algorithm out of the above theoretical results. In particular, we propose to approximate the flows $F$ such that the flow consistency equations are respected at convergence with enough capacity in our estimator of $F$, just like the Bellman equations for temporal-difference (TD) algorithms (Sutton and Barto, 2018). This could yield the following objective for a trajectory $\tau$:

$$\tilde{\mathcal{L}}_\theta(\tau) = \sum_{s' \in \tau \neq s_0} \left( \sum_{s,a:T(s,a)=s'} F_\theta(s,a) - R(s') - \sum_{a' \in \mathcal{A}(s')} F_\theta(s',a') \right)^2. \tag{11}$$

One issue from a learning point of view is that the flow will be very large for nodes near the root (early in the trajectory) and tiny for nodes near the leaves (late in the trajectory). In high-dimensional spaces where the cardinality of $\mathcal{X}$ is exponential (e.g., in the typical number of actions to form an $x$), the $F(s, a)$ and $F(s)$ for early states will be exponentially larger than for later states. Since we want $F(s, a)$ to be the output of a neural network, this would lead to serious numerical issues.

To avoid this problem, we define the flow matching objective on a log-scale, where we match not the incoming and outgoing flows but their logarithms, and we train our predictor to estimate $F_\theta^{\log}(s, a) = \log F(s, a)$, and exponentiate-sum-log the $F_\theta^{\log}$ predictions to compute the loss, yielding the square of a difference of logs:

$$\mathcal{L}_{\theta,\epsilon}(\tau) = \sum_{s' \in \tau \neq s_0} \left( \log \left[ \epsilon + \sum_{s,a:T(s,a)=s'} \exp F_\theta^{\log}(s, a) \right] - \log \left[ \epsilon + R(s') + \sum_{a' \in \mathcal{A}(s')} \exp F_\theta^{\log}(s', a') \right] \right)^2 \quad (12)$$

which gives equal gradient weighing to large and small magnitude predictions. Note that matching the logs of the flows is equivalent to making the ratio of the incoming and outgoing flow closer to 1. To give more weight to errors on large flows and avoid taking the logarithm of a tiny number, we compare $\log(\epsilon+\text{incoming flow})$ with $\log(\epsilon+\text{outgoing flow})$. It does not change the global minimum, which is still when the flow equations are satisfied, but it avoids numerical issues with taking the log of a tiny flow. The hyper-parameter $\epsilon$ also trades-off how much pressure we put on matching large versus small flows, and in our experiments is set to be close to the smallest value $R$ can take. Since we want to discover the top modes of $R$, it makes sense to care more for the larger flows. Many other objectives are possible for which flow matching is also a global minimum.

An interesting advantage of such objective functions is that they yield off-policy offline methods. The predicted flows $F$ do not depend on the policy used to sample trajectories (apart from the fact that the samples should sufficiently cover the space of trajectories in order to obtain generalization). This is formalized below, which shows that we can use any broad-support policy to sample training trajectories and still obtain the correct flows and generative model, i.e., training can be off-policy.

**Proposition 3.** *Let trajectories $\tau$ used to train $F_\theta$ be sampled from an exploratory policy $P$ with the same support as the optimal $\pi$ defined in Eq. 5 for a consistent flow $F^* \in \mathcal{F}^*$. A flow is consistent if Eq. 4 is respected. Also assume that $\exists \theta : F_\theta = F^*$, i.e., we choose a sufficiently rich family of predictors. Let $\theta^* \in \operatorname{argmin}_\theta E_{P(\tau)}[L_\theta(\tau)]$ a minimizer of the expected training loss. Let $L_\theta(\tau)$ have the property that when flows are matched it achieves its lowest possible value. First, it can be shown that this property is satisfied for the loss in Eq. 12. Then*

$$F_{\theta^*} = F^*, \quad \text{and} \quad L_{\theta^*}(\tau) = 0 \quad \forall \tau \sim P(\theta), \quad (13)$$

*i.e., a global optimum of the expected loss provides the correct flows. If $\pi_{\theta^*}(a|s) = \frac{F_{\theta^*}(s,a)}{\sum_{a' \in \mathcal{A}(s)} F_{\theta^*}(s,a')}$ then we also have*

$$\pi_{\theta^*}(x) = \frac{R(x)}{Z}. \quad (14)$$

The proof is in Appendix A.1. Note that, in RL terms, this method is akin to asynchronous dynamic programming (Sutton and Barto, 2018, §4.5), which is an off-policy off-line method which converges provided every state is visited infinitely many times asymptotically.

## 3 Related Work

The objective of training a policy generating states with a probability proportional to rewards was presented by Buesing et al. (2019) but the proposed method only makes sense when there is a bijection between action sequences and states. In contrast, GFlowNet is applicable in the more general setting where many paths can lead to the same state. The objective to sample with probability proportional to a given unnormalized positive function is achieved by many MCMC methods (Grathwohl et al., 2021; Dai et al., 2020). However, when mixing between modes is challenging (e.g., in high-dimensional spaces with well-separated modes occupying a fraction of the total volume) convergence to the target distribution can be extremely slow. In contrast, GFlowNet is not iterative and amortizes the challenge of sampling from such modes through a training procedure which must be sufficiently exploratory.

This sampling problem comes up in molecule generation and has been studied in this context with numerous generative models (Shi et al., 2020; Jin et al., 2020; Luo et al., 2021), MCMC methods (Seff

et al., 2019; Xie et al., 2021), RL (Segler et al., 2017; Cao and Kipf, 2018; Popova et al., 2019; Gottipati et al., 2020; Angermueller et al., 2020) and evolutionary methods (Brown et al., 2004; Jensen, 2019; Swersky et al., 2020). Some of these methods rely on a given set of "positive examples" (high-reward) to train a generative model, thus not taking advantage of the "negative examples" and the continuous nature of the measurements (some examples should be generated more often than others). Others rely on the traditional return maximization objectives of RL, which tends to focus on one or a few dominant modes, as we find in our experiments. Beyond molecules, there are previous works generating data non-greedily through RL (Bachman and Precup, 2015) or energy-based GANs (Dai et al., 2017).

The objective that we formulate in (12) may remind the reader of the objective of control-as-inference's Soft Q-Learning (Haarnoja et al., 2017), with the difference that we include *all* the parents of a state in the in-flow, whereas Soft Q-Learning only uses the parent contained in the trajectory. Soft Q-Learning induces a different policy, as shown by Proposition 1, one where $P(\tau) \propto R(\tau)$ rather than $P(x) \propto R(x)$. More generally, we only consider deterministic generative settings whereas RL is a more general framework for stochastic environments.

Literature at the intersection of network flow and deep learning is sparse, and is mostly concerned with solving maximum flow problems (Nazemi and Omidi, 2012; Chen and Zhang, 2020) or classification within existing flow networks (Rahul et al., 2017; Pektaş and Acarman, 2019). Finally, the idea of accounting for the search space being a DAG rather than a tree in MCTS, known as transpositions (Childs et al., 2008), also has some links with the proposed method.

## 4 Empirical Results

We first verify that GFlowNet works as advertised on an artificial domain small enough to compute the partition function exactly, and compare its abilities to recover modes compared to standard MCMC and RL methods, with its sampling distribution better matching the normalized reward. We find that GFlowNet (A) converges to $\pi(x) \propto R(x)$, (B) requires less samples to achieve some level of performance than MCMC and PPO methods and (C) recovers all the modes and does so faster than MCMC and PPO, both in terms of wall-time and number of states visited and queried. We then test GFlowNet on a large scale domain, which consists in generating small drug molecule graphs, with a reward that estimates their binding affinity to a target protein (see Appendix A.3). We find that GFlowNet finds higher reward and more diverse molecules faster than baselines.

### 4.1 A (hyper-)grid domain

Consider an MDP where states are the cells of a $n$-dimensional hypercubic grid of side length $H$. The agent starts at coordinate $x = (0, 0, ...)$ and is only allowed to increase coordinate $i$ with action $a_i$ (up to $H$, upon which the episode terminates). A *stop* action indicates to terminate the trajectory. There are many action sequences that lead to the same coordinate, making this MDP a DAG. The reward for ending the trajectory in $x$ is some $R(x) > 0$. For MCMC methods, in order to have an ergodic chain, we allow the iteration to decrease coordinates as well, and there is no *stop* action.

We ran experiments with this reward function:

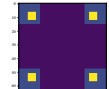

$$R(x) = R_0 + R_1 \prod_i \mathbb{I}(0.25 < |x_i/H - 0.5|) + R_2 \prod_i \mathbb{I}(0.3 < |x_i/H - 0.5| < 0.4)$$

with $0 < R_0 \ll R_1 < R_2$, pictured when $n = 2$ on the right. For this choice of $R$, there are only interesting rewards near the corners of the grid, and there are exactly $2^n$ modes. We set $R_1 = 1/2$, $R_2 = 2$. By varying $R_0$ and setting it closer to 0, we make this problem artificially harder, creating a region of the state space which it is undesirable to explore. To measure the performance of a method, we measure the empirical L1 error $\mathbb{E}[|p(x) - \pi(x)|]$. $p(x) = R(x)/Z$ is known in this domain, and $\pi$ is estimated by repeated sampling and counting frequencies for each possible $x$. We also measure the number of modes with at least 1 visit as a function of the number of states visited.

We run the above experiment for $R_0 \in \{10^{-1}, 10^{-2}, 10^{-3}\}$ with $n = 4$, $H = 8$. In Fig. 2 we see that GFlowNet is robust to $R_0$ and obtains a low L1 error, while a Metropolis-Hastings-MCMC based method requires exponentially more samples than GFlowNet to achieve some level of L1 error. This is apparent in Fig. 2 (with a log-scale horizontal axis) by comparing the slope of progress of GFlowNet (beyond the initial stage) and that of the MCMC sampler. We also see that MCMC takes much longer to visit each mode *once* as $R_0$ decreases, while GFlowNet is only slightly affected, with GFlowNet converging to some level of L1 error faster, as per hypothesis (B). This suggests that

GFlowNet is robust to the separation between modes (represented by $R_0$ being smaller) and thus recovers all the modes much faster than MCMC (again, noting the log-scale of the horizontal axis).

To compare to RL, we run PPO (Schulman et al., 2017). To discover all the modes in a reasonable time, we need to set the entropy maximization term much higher (0.5) than usual ($\ll 1$). We verify that PPO is not overly regularized by comparing it to a random agent. PPO finds all the modes faster than uniform sampling, but much more slowly than GFlowNet, and is also robust to the choice of $R_0$. This and the previous result validates hypothesis (C). We also run SAC (Haarnoja et al., 2018), finding similar or worse results. We provide additional results and discussion in Appendix A.6.

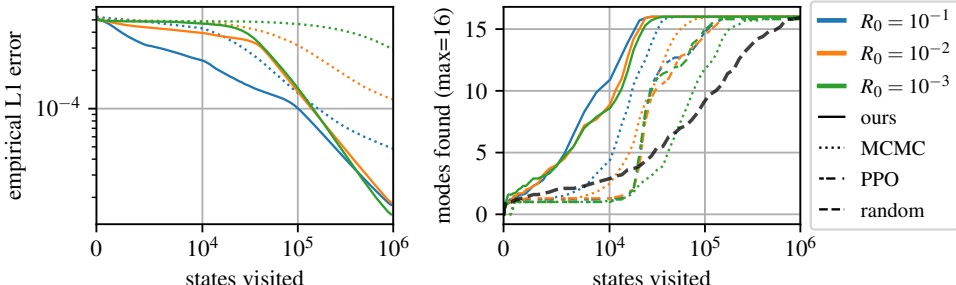

Figure 2: Hypergrid domain. Changing the task difficulty $R_0$ to illustrate the advantage of GFlowNet over others. We see that as $R_0$ gets smaller, MCMC struggles to fit the distribution because it struggles to visit all the modes. PPO also struggles to find all the modes, and requires very large entropy regularization, but is robust to the choice of $R_0$. We plot means over 10 runs for each setting.

## 4.2 Generating small molecules

Here our goal is to generate a diverse set of small molecules that have a high reward. We define a large-scale environment which allows an agent to sequentially generate molecules. This environment is challenging, with up to $10^{16}$ states and between 100 and 2000 actions depending on the state.

We follow the framework of Jin et al. (2020) and generate molecules by parts using a predefined vocabulary of building blocks that can be joined together forming a *junction tree* (detailed in A.3). This is also known as fragment-based drug design (Kumar et al., 2012; Xie et al., 2021). Generating such a graph can be described as a sequence of additive edits: given a molecule and constraints of chemical validity, we choose an atom to attach a block to. The action space is thus the product of choosing where to attach a block and choosing which block to attach. There is an extra action to stop the editing sequence. This sequence of edits yields a DAG MDP, as there are multiple action sequences that lead to the same molecule graph, and no edge removal actions, which prevents cycles.

The reward is computed with a pretrained *proxy* model that predicts the binding energy of a molecule to a particular protein target (soluble epoxide hydrolase, sEH, see A.3). Although computing binding energy is computationally expensive, we can call this proxy cheaply. Note that for realistic drug design, we would need to consider many more quantities such as drug-likeness (Bickerton et al., 2012), toxicity, or synthesizability. Our goal here is not solve this problem, and our work situates itself within such a larger project. Instead, we want to show that given a proxy $R$ in the space of molecules, we can quickly match its induced distribution $\pi(x) \propto R(x)$ and find many of its modes.

We parameterize the proxy with an MPNN (Gilmer et al., 2017) over the atom graph. Our flow predictor $F_\theta$ is parameterized similarly to MARS (Xie et al., 2021), with an MPNN, but over the junction tree graph (the graph of blocks), which had better performance. For fairness, this architecture is used for both GFlowNet and the baselines. Complete details can be found in Appendix A.4.

We pretrain the proxy with a semi-curated semi-random dataset of 300k molecules (see A.4) down to a test MSE of 0.6; molecules are scored according to the docking score (Trott and Olson, 2010), renormalized so that most scores fall between 0 and 10 (to have $R(x) > 0$). We plot the dataset's reward distribution in Fig. 3. We train all generative models with up to $10^6$ molecules. During training, sampling follows exploratory policy $P(a|s)$ which is a mixture between $\pi(a|s)$ (Eq. 5), used with probability 0.95, and a uniform distribution over allowed actions with probability 0.05.

**Experimental results** In Fig. 3 we show the empirical distribution of rewards in two settings; first when we train our model with $R(x)$, then with $R(x)^\beta$. If GFlowNet learns a reasonable policy $\pi$,

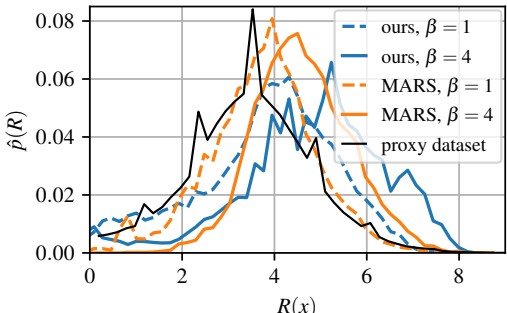
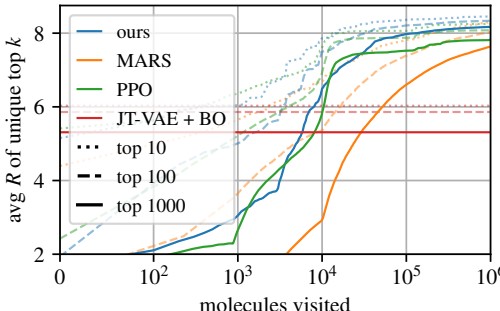

Figure 3: Empirical density of rewards. We verify that GFlowNet is consistent by training it with $R^\beta$, $\beta = 4$, which has the hypothesized effect of shifting the density to the right.

Figure 4: The average reward of the top-$k$ as a function of learning (averaged over 3 runs). Only unique hits are counted. Note the log scale. Our method finds more unique good molecules faster.

this should shift the distribution to the right. This is indeed what we observe. We compare GFlowNet to MARS (Xie et al., 2021), known to work well in the molecule domain, and observe the same shift. Note that GFlowNet finds more high reward molecules than MARS with these $\beta$ values; this is consistent with the hypothesis that it finds high-reward modes faster (since MARS is an MCMC method, it would eventually converge to the same distribution, but takes more time).

In Fig. 4, we show the average reward of the top-$k$ molecules found so far, without allowing for duplicates (based on SMILES). We compare GFlowNet with MARS, PPO, and JT-VAE (Jin et al., 2020) with Bayesian Optimization. As expected, PPO plateaus after a while; RL tends to be satisfied with good enough trajectories unless it is strongly regularized with exploration mechanisms. For GFlowNet and for MARS, the more molecules are visited, the better they become, with a slow convergence towards the proxy's max reward. Given the same compute time, JT-VAE+BO generates only about $10^3$ molecules (due to its expensive Gaussian Process) and so does not perform well.

The maximum reward in the proxy's dataset is 10, with only 233 examples above 8. In our best run, we find 2339 unique molecules during training with a score above 8, only 39 of which are in the dataset. We compute the average pairwise Tanimoto similarity for the top 1000 samples: GFlowNet has a mean of $0.44 \pm 0.01$, PPO, $0.62 \pm 0.03$, and MARS, $0.59 \pm 0.02$ (mean and std over 3 runs). As expected, our MCMC baseline (MARS) and RL baseline (PPO) find less diverse candidates. We also find that GFlowNet discovers **many** more modes ($>1500$ with $R>8$ vs $<100$ for MARS). This is shown in Fig. 5 where we consider a mode to be a Bemis-Murcko scaffold (Bemis and Murcko, 1996), counted for molecules above a certain reward threshold. We provide additional insights into how GFlowNet matches the rewards in Appendix A.7.

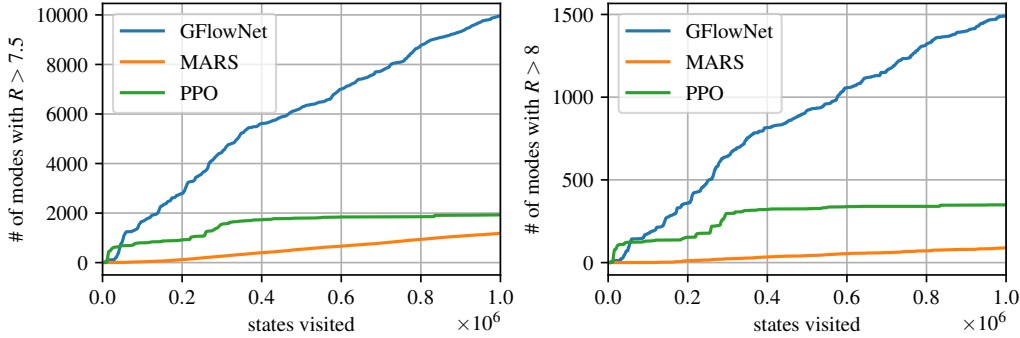

Figure 5: Number of diverse Bemis-Murcko scaffolds found above reward threshold $T$ as a function of the number of molecules seen. Left, $T = 7.5$. Right, $T = 8$.

## 4.3 Multi-Round Experiments

To demonstrate the importance of diverse candidate generation in an active learning setting, we consider a sequential acquisition task. We simulate the setting where there is a limited budget for calls to the true oracle $O$. We use a proxy $M$ initialized by training on a limited dataset of $(x, R(x))$

pairs $D_0$, where $R(x)$ is the true reward from the oracle. The generative model ($\pi_\theta$) is trained to fit to the unnormalized probability function learned by the proxy $M$. We then sample a batch $B = \{x_1, x_2, \dots x_k\}$ where $x_i \sim \pi_\theta$, which is evaluated with the oracle $O$. The proxy $M$ is updated with this newly acquired and labeled batch, and the process is repeated for $N$ iterations. We discuss the experimental setting in more detail in Appendix A.5.

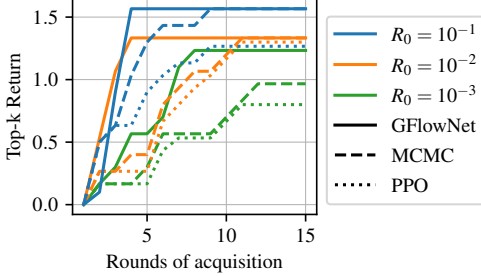
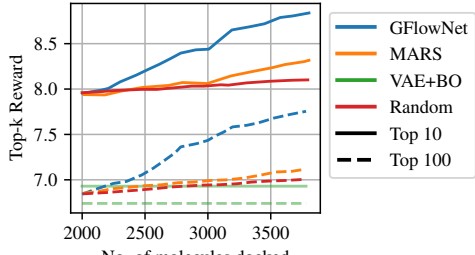

Figure 6: The top-k return (mean over 3 runs) in the 4-D Hyper-grid task with active learning. GFlowNet gets the highest return faster.

Figure 7: The top-k docking reward (mean over 3 runs) in the molecule task with active learning. GFlowNet consistently generates better samples.

**Hyper-grid domain** We present results for the multi-round task in the 4-D hyper-grid domain in Figure 6. We use a Gaussian Process (Williams and Rasmussen, 1995) as the proxy. We compare the *Top-k Return* for all the methods, which is defined as $\text{mean}(\text{top-}k(D_i)) - \text{mean}(\text{top-}k(D_{i-1}))$, where $D_i$ is the dataset of points acquired until step $i$, and $k = 10$ for this experiment. The initial dataset $D_0$ ($|D_0| = 512$) is the same for all the methods compared. We observe that GFlowNet consistently outperforms the baselines in terms of return over the initial set. We also observe that the mean pairwise L2-distance between the $\text{top-}k$ points at the end of the final round is $0.83 \pm 0.03$, $0.61 \pm 0.01$ and $0.51 \pm 0.02$ for GFlowNet, MCMC and PPO respectively. This demonstrates the ability of GFlowNet to capture the modes, even in the absence of the true oracle, as well as the importance of capturing this diversity in multi-round settings.

**Small Molecules** For the molecule discovery task, we initialize an MPNN proxy to predict docking scores from AutoDock (Trott and Olson, 2010), with $|D_0| = 2000$ molecules. At the end of each round we generate 200 molecules which are evaluated with AutoDock and used to update the proxy. Figure 7 shows GFlowNet discovers molecules with significantly higher energies than the initial set $D_0$. It also consistently outperforms MARS as well as Random Acquisition. PPO training was unstable and diverged consistently so the numbers are not reported. The mean pairwise Tanimoto similarity in the initial set is 0.60. At the end of the final round, it is $0.54 \pm 0.04$ for GFlowNet and $0.64 \pm 0.03$ for MARS. This further demonstrates the ability of GFlowNet to generate diverse candidates, which ultimately helps improve the final performance on the task. Similar to the single step setting, we observe that JT-VAE+BO is only able to generate $10^3$ molecules with similar compute time, and thus performs poorly.

## 5    Discussion & Limitations

In this paper we have introduced a novel TD-like objective for learning a flow for each state and *(state, action)* pair such that policies sampling actions proportional to these flows draw terminal states in proportion to their reward. This can be seen as an alternative approach to turn an energy function into a fast generative model, without the need for an iterative method like that needed with MCMC methods, and with the advantage that when training succeeds, the policy generates a great diversity of samples near the main modes of the target distribution without being slowed by issues of mixing between modes.

**Limitations.** One downside of the proposed method is that, as for TD-based methods, the use of bootstrapping may cause optimization challenges (Kumar et al., 2020; Bengio et al., 2020) and limit its performance. In applications like drug discovery, sampling from the regions surrounding each mode is already an important advantage, but future work should investigate how to combine such a generative approach to local optimization in order to refine the generated samples and approach the local maxima of reward while keeping the batches of candidates diverse.

**Negative Social Impact.** The authors do not foresee negative social impacts of this work specifically.

## Acknowledgments and Disclosure of Funding

This research was enabled in part by computational resources provided by Calcul Québec (`www.calculquebec.ca`) and Compute Canada (`www.computecanada.ca`). All authors are funded by their primary academic institution. We also acknowledge funding from Samsung Electronics Co., Ldt., CIFAR and IBM.

The authors are grateful to Andrei Nica for generating the molecule dataset, to Maria Kadukova for advice on molecular docking, to Harsh Satija for feedback on the paper, as well as to all the members of the Mila Molecule Discovery team for the many research discussions on the challenges we faced.

## Author Contributions

EB and YB contributed to the original idea, and wrote most sections of the paper. YB wrote the proofs of Propositions 1-3, EB the proof of Proposition 4. EB wrote the code and ran experiments for sections 4.1 (hypergrid) and 4.2 (small molecules). MJ wrote the code and ran experiments for section 4.3 (multi-round) and wrote the corresponding results section of the paper. MK wrote the biochemical framework upon which the molecule experiments are built, assisted in debugging and running experiments for section 4.3, implemented mode-counting routines used in 4.2, and wrote the biochemical details of the paper.

MK, DP and YB provided supervision for the project. All authors contributed to proofreading and editing the paper.

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
