# A Appendix

All our ML code uses the PyTorch (Paszke et al., 2019) library. We reimplement RL and other baselines. We use the AutoDock Vina (Trott and Olson, 2010) library for binding energy estimation and RDKit (Landrum) for chemistry routines.

Running all the molecule experiments presented in this paper takes an estimated 26 GPU days. We use a cluster with NVidia V100 GPUs. The grid experiments take an estimated 8 CPU days (for a single-core).

All implementations are available at https://github.com/bengioe/gflownet.

## A.1 Proofs

**Proposition 1.** *Let $C : \mathcal{A}^* \mapsto \mathcal{S}$ associate each allowed action sequence $\vec{a} \in \mathcal{A}^*$ to a state $s = C(\vec{a}) \in \mathcal{S}$. Let $\tilde{V} : \mathcal{S} \mapsto \mathbf{R}^+$ associate each state $s \in \mathcal{S}$ to $\tilde{V}(s) = \sum_{\vec{b} \in \mathcal{A}^*(s)} R(s + \vec{b}) > 0$, where $\mathcal{A}^*(s)$ is the set of allowed continuations from $s$ and $s + \vec{b}$ denotes the resulting state, i.e., $\tilde{V}(s)$ is the sum of the rewards of all the states reachable from $s$. Consider a policy $\pi$ which starts from the state corresponding to the empty string $s_0 = C(\emptyset)$ and chooses from state $s \in \mathcal{S}$ an allowable action $a \in \mathcal{A}(s)$ with probability $\pi(a|s) = \frac{\tilde{V}(s+a)}{\sum_{b \in \mathcal{A}(s)} \tilde{V}(s+b)}$. Denote $\pi(\vec{a} = (a_1, \ldots, a_N)) = \prod_{i=1}^N \pi(a_i | C(a_1, \ldots, a_{i-1}))$ and $\pi(s)$ with $s \in \mathcal{S}$ the probability of visiting a state $s$ with this policy. The following then obtains:*
*(a) $\pi(s) = \sum_{\vec{a}_i : C(\vec{a}_i) = s} \pi(\vec{a}_i)$.*
*(b) If $C$ is bijective, then $\pi(s) = \frac{\tilde{V}(s)}{\tilde{V}(s_0)}$ and as a special case for terminal states $x$, $\pi(x) = \frac{R(x)}{\sum_{x \in \mathcal{X}} R(x)}$.*
*(c) If $C$ is non-injective and there are $n(x)$ distinct action sequences $\vec{a}_i$ s.t. $C(\vec{a}_i) = x$, then $\pi(x) = \frac{n(x)R(x)}{\sum_{x' \in \mathcal{X}} n(x')R(x')}$.*

*Proof.* Since $s$ can be reached (from $s_0$) according to any of the action sequences $\vec{a}_i$ such that $C(\vec{a}_i) = s$ and they are mutually exclusive and cover all the possible ways of reaching $s$, the probability that $\pi$ visits state $s$ is simply $\sum_{\vec{a}_i : C(\vec{a}_i) = s} \pi(\vec{a}_i)$, i.e., we obtain (a).

If $C$ is bijective, it means that there is only one such action sequence $\vec{a} = (a_1, \ldots, a_N)$ landing in state $s$, and the set of action sequences and states forms a tree rooted at $s_0$. hence by (a) we get that $\pi(s) = \pi(\vec{a})$. First note that because $\tilde{V}(s) = \sum_{\vec{b} \in \mathcal{A}^*(s)} R(s + \vec{b})$, i.e., $\tilde{V}(s)$ is the sum of the terminal rewards for all the leaves rooted at $s$, we have that $\tilde{V}(s) = \sum_{b \in \mathcal{A}(s)} V(s + b)$. Let us now prove by induction that $\pi(s) = \frac{\tilde{V}(s)}{\tilde{V}(s_0)}$. It is true for $s = s_0$ since $\pi(s_0 = 1)$ (i.e., every trajectory includes $s_0$). Assuming it is true for $s' = C(a_1, \ldots, a_{N-1})$, consider $s = C(a_1, \ldots, a_N)$:

$$\pi(s) = \pi(a_N | s')\pi(s') = \frac{\tilde{V}(s)}{\sum_{b \in \mathcal{A}(s')} \tilde{V}(s' + b)} \frac{\tilde{V}(s')}{\tilde{V}(s_0)}.$$

Using our above result that $\tilde{V}(s) = \sum_{b \in \mathcal{A}(s)} \tilde{V}(s + b)$, we thus obtain a cancellation of $\tilde{V}(s')$ with $\sum_{b \in \mathcal{A}(s')} \tilde{V}(s' + b)$ and obtain

$$\pi(s) = \frac{\tilde{V}(s)}{\tilde{V}(s_0)}, \tag{15}$$

proving that the recursion holds. We already know from the definition of $\tilde{V}$ that $\tilde{V}(s_0) = \sum_{x \in \mathcal{X}} R(x)$, so for the special case of $x$ a terminal state, $\tilde{V}(x) = R(x)$ and Eq. 15 becomes $\pi(x) = \frac{R(x)}{\sum_{x' \in \mathcal{X}} R(x')}$, which finishes to prove (b).

On the other hand, if $C$ is non-injective, the set of paths forms a DAG, and not a tree. Let us transform the DAG into a tree by creating a new state-space (for the tree version) which is the action sequence itself. Note how the same original leaf node $x$ is now repeated $n(x)$ times in the tree (with leaves denoted by action sequences $\vec{a}_i$) if there are $n(x)$ action sequences leading to $x$ in the DAG. With the same definition of $\tilde{V}$ and $\pi(a|s)$ but in the tree, we obtain all the results from (b) (which are applicable

because we have a tree), and in particular $\pi(\vec{a}_i)$ under the tree is proportional to $R(x') = R(x)$. Applying (a), we see that $\pi(x) \propto n(x)R(x)$, which proves (c). $\qquad\square$

**Proposition 3.** *Let trajectories $\tau$ used to train $F_\theta$ be sampled from an exploratory policy $P$ with the same support as the optimal $\pi$ defined in Eq. 5 for a consistent flow $F^* \in \mathcal{F}^*$. A flow is consistent if Eq. 4 is respected. Also assume that $\exists \theta : F_\theta = F^*$, i.e., we choose a sufficiently rich family of predictors. Let $\theta^* \in \arg\min_\theta E_{P(\tau)}[L_\theta(\tau)]$ a minimizer of the expected training loss. Let $L_\theta(\tau)$ have the property that when flows are matched it achieves its lowest possible value. First, it can be shown that this property is satisfied for the loss in Eq. 12. Then*

$$F_{\theta^*} = F^*, \quad \text{and} \tag{16}$$
$$L_{\theta^*}(\tau) = 0 \quad \forall \tau \sim P(\theta), \tag{17}$$

*i.e., a global optimum of the expected loss provides the correct flows. If*

$$\pi_{\theta^*}(a|s) = \frac{F_{\theta^*}(s, a)}{\sum_{a' \in \mathcal{A}(s)} F_{\theta^*}(s, a')} \tag{18}$$

*then we also have*

$$\pi_{\theta^*}(x) = \frac{R(x)}{Z}. \tag{19}$$

*Proof.* A per-trajectory loss of 0 can be achieved by choosing a $\theta$ such that $F_\theta = F^*$ (which we assumed was possible), since this makes the incoming flow equal the outgoing flow. Note that there always exists a solution $F^*$ in the space of allow possible flow functions which satisfies the flow equations (incoming = outgoing) by construction of flow networks with only a constraint on the flow in the terminal nodes (leaves). Since having $L_\theta(\tau)$ equal to 0 for all $\tau \sim P(\theta)$ makes the expected loss 0, and this is the lowest achievable value (since $L_\theta(\tau) \geq 0 \ \forall \theta$), it means that such a $\theta$ is a global minimizer of the expected loss, and we can denote it $\theta^*$. If we optimize $F$ in function space, we can directly set to 0 the gradient of the loss with respect to $F(s, a)$ separately, and find a solution.

Since we have chosen $P$ with support large enough to include all the trajectories leading to a terminal state $R(x) > 0$, it means that $L_\theta(\tau) = 0$ for all these trajectories and that $F_\theta = F^*$ for all nodes on these trajectories. We can then apply Proposition 2 (since the flows match everywhere and we have defined the policy correspondingly, as per Eq. 5). We then obtain the conclusion by applying result (c) from Proposition 2. $\qquad\square$

Note that in the general case, an infinite number of solutions exist. Consider the case where two trajectories are possible, say $s_0, a_1, s_A, a_2, s_T$ and $s_0, a_3, s_B, a_4, s_T$, and both lead to the same terminal state $s_T$ with reward $r$. Then a valid solution solves the constrained system of equations $F(s_A) + F(s_B) = r, F(s_A) > 0, F(s_B) > 0$, and we see that there is an infinite number of solutions described by one parameter $u$ where $F(s_A) = u, F(s_B) = r - u \ u \in [0, r]$.

## A.2 Action-value function equivalence

Here we show that the flow $F(s, a)$ that the proposed method learns can correspond to a "real" action-value function $\hat{Q}^\mu(s, a)$ in an RL sense, for a policy $\mu$.

First note that this is in a way trivially true: in inverse RL (Ng et al., 2000) there typically exists an infinite number of solutions to defining $\hat{R}$ from a policy $\pi$ such that $\pi = \arg\max_{\pi_i} V^{\pi_i}(s; \hat{R}) \ \forall s$, where $V^{\pi_i}(s; \hat{R})$ is the value function at $s$ for reward function $\mathbf{R}$.

More interesting is the case where $F(s, a; R)$ obtained from computing the flow corresponding to $R$ is exactly equal to some $Q^\mu(s, a; \hat{R})$ modulo a multiplicative factor $f(s)$. What are $\mu$ and $\hat{R}$?

In the bijective case a simple answer exists.

**Proposition 4.** *Let $\mu$ be the uniform policy such that $\mu(a|s) = 1/|\mathcal{A}(s)|$, let $f(x) = \prod_{t=0}^n |\mathcal{A}(s_t)|$ when $x \equiv (s_0, s_1, ..., s_n)$, and let $\hat{R}(x) = R(x)f(s_{n-1})$, then $Q^\mu(s, a; \hat{R}) = F(s, a; R)f(s)$.*

*Proof.* By definition of the action-value function in terms of the action-value at the next step and by definition of $\mu$:

$$Q^\mu(s, a; \hat{R}) = \hat{R}(s') + \frac{1}{|\mathcal{A}(s')|} \sum_{a' \in \mathcal{A}(s')} Q^\mu(s', a'; \hat{R}) \tag{20}$$

where $s' = T(s, a)$, as the environment is deterministic and has a tree structure.

For some leaf $s'$, $Q^\mu(s, a; \hat{R}) = \hat{R}(s') = R(s')f(s)$. Again for some leaf $s'$, the flow is $F(s, a; R) = R(s')$. Thus $Q^\mu(s, a; \hat{R}) = F(s, a; R)f(s)$. Suppose (20) is true, then by induction for a non-leaf $s'$:

$$Q^\mu(s, a; \hat{R}) = \hat{R}(s') + \frac{1}{|\mathcal{A}(s')|} \sum_{a' \in \mathcal{A}(s')} Q^\mu(s', a'; \hat{R}) \tag{21}$$

$$Q^\mu(s, a; \hat{R}) = 0 + \frac{1}{|\mathcal{A}(s')|} \sum_{a' \in \mathcal{A}(s')} F(s', a'; R)f(s') \tag{22}$$

we know from Eq 4 that

$$F(s, a; R) = \sum_{a' \in \mathcal{A}(s')} F(s', a'; R) \tag{23}$$

and since $f(s') = f(s)|\mathcal{A}(s')|$, we have that:

$$Q^\mu(s, a; \hat{R}) = \frac{F(s, a; R)f(s')}{|\mathcal{A}(s')|} \tag{24}$$

$$= \frac{F(s, a; R)f(s)|\mathcal{A}(s')|}{|\mathcal{A}(s')|} \tag{25}$$

$$= F(s, a; R)f(s) \tag{26}$$

$\square$

Thus we have shown that the flow in a bijective case corresponds to the action-value of the uniform policy. This result suggests that the policy evaluation of the uniform policy learns something non-trivial in the tree MDP case. Perhaps such a quantity could be used in other interesting ways.

In the non-injective case, since an infinite number of valid flows exists, it's not clear that such a simple equivalence always exists.

As a particular case, consider the flow $F$ which assigns exactly 0 flow to edges that would induce multiple paths to any node. In other words, consider the flow which induces a tree, i.e. a bijection between action sequences and states, by disallowing flow between edges not in that bijection. By Proposition 4, we can recover some valid $Q^\mu$.

Since there is at least one flow for which this equivalence exists, we conjecture that more general mappings between flows and action-value functions exist.

**Conjecture** *There exists $f$ a function of $n(s)$ the number of paths to $s$, $\mathcal{A}(s)$, and $n_p(s) = |\{(p, a)|T(p, a) = s\}|$ the number of parents of $s$, such that $f(s, n(s), n_p(s), \mathcal{A}(s))Q^\mu(s, a; \hat{R}) = F(s, a; R)$ and $\hat{R}(x) = R(x)f(x)$ for the uniform policy $\mu$ and for some valid flow $F(s, a; R)$.*

### A.3 Molecule domain details

We allow the agent to choose from a library of 72 predefined blocks. We duplicate blocks from the point of view of the agent to allow attaching to different symmetry groups of a given block. This yields a total of 105 actions per stem; stems are atoms where new blocks can be attached to. We choose the blocks via the process suggested by Jin et al. (2020) over the ZINC dataset (Sterling and Irwin, 2015). We allow the agent to generate up to 8 blocks.

The 72 block SMILES are Br, C, C#N, C1=CCCCC1, C1=CNC=CC1, C1CC1, C1CCCC1, C1CCCCC1, C1CCNC1, C1CCNCC1, C1CCOC1, C1CCOCC1, C1CNCCN1, C1COCCN1, C1COCC[NH2+]1, C=C, C=C(C)C, C=CC, C=N, C=O, CC, CC(C)C, CC(C)O, CC(N)=O, CC=O, CCC, CCO, CN, CNC, CNC(C)=O, CNC=O, CO, CS, C[NH3+], C[SH2+], Cl, F, FC(F)F, I, N, N=CN, NC=O, N[SH](=O)=O, O, O=CNO, O=CO, O=C[O-], O=PO, O=P[O-], O=S=O, O=[NH+][O-],

```
O=[PH](O)O, O=[PH]([O-])O, O=[SH](=O)O, O=[SH](=O)[O-], O=c1[nH]cnc2[nH]cnc12,
O=c1[nH]cnc2c1NCCN2, O=c1cc[nH]c(=O)[nH]1, O=c1nc2[nH]c3ccccc3nc-2c(=O)[nH]1,
O=c1nccc[nH]1, S, c1cc[nH+]cc1, c1cc[nH]c1, c1ccc2[nH]ccc2c1, c1ccc2ccccc2c1,
c1ccccc1, c1ccncc1, c1ccsc1, c1cn[nH]c1, c1cncnc1, c1cscn1, c1ncc2nc[nH]c2n1.
```

We illustrate these building blocks and their attachment points in Figure 8.

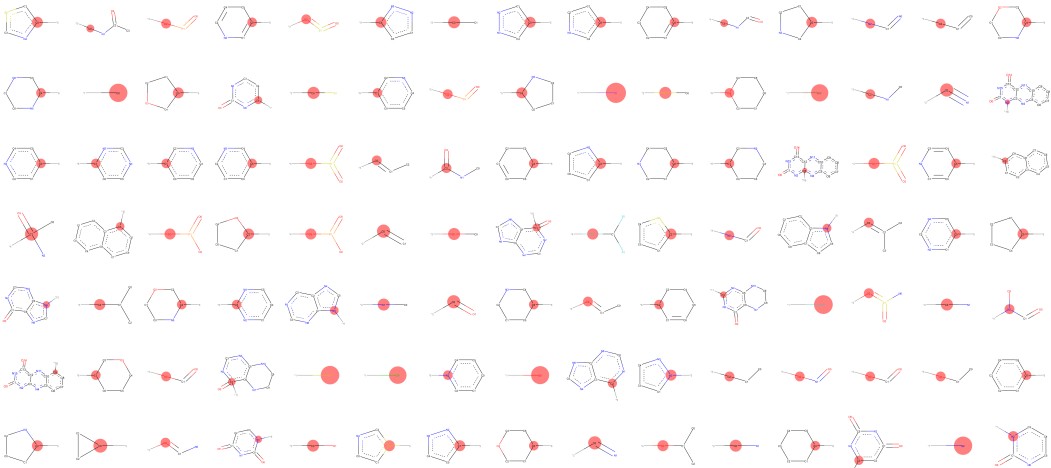

Figure 8: The list of building blocks used in molecule design. The stem, the atom which connects the block to the rest of the molecule, is highlighted.

We compute the reward based on a proxy's prediction. This proxy is trained on a dataset of 300k randomly generated molecules, whose binding affinity with a target protein has been computed with AutoDock (Trott and Olson, 2010). Since the binding affinity is an energy where lower is better, we takes its opposite and then renormalize it (subtract the mean, divide by the standard deviation) to obtain the reward.

We use the sEH protein and its 4JNC inhibitor. The soluble epoxide hydrolase, or sEH, is a well studied protein which plays a role in respiratory and heart disease, which makes it an interesting pharmacological target and benchmark for ML methods.

Note that we also experimented with other biologically relevant quantities, in particular logP (the n-octanol-water partition coefficient) and QED (Bickerton et al., 2012). Both were very easy to maximize with GFlowNet. For logP we quickly find molecules with a >20 logP, which at this point is biologically uninteresting (for reference, ibuprofen's logP is between 3.5 and 4). For QED, we also quickly find molecules with the maximum possible QED in our action space, which is 0.948 (in fact our top-1000 is >0.94 after 100k molecules seen). Since docking is a much harder oracle, we focused on it. Also note that we experimented with combining different scores multiplicatively (e.g. multiplying docking score by a renormalized QED and synthesizability), with some success. A more specific contribution in that regards is left to future work.

### A.4 Molecule domain implementation details

For the proxy of the oracle, from which the reward is defined, we use an MPNN (Gilmer et al., 2017) that receives the atom graph as input. We compute the atom graph using RDKit. Each node in the graph has features including the one-hot vector of its atomic number, its hybridization type, its number of implicit hydrogens, and a binary indicator of it being an acceptor or a donor atom. The MPNN uses a GRU at each iteration as the graph convolution layer is applied iteratively for 12 steps, followed by a Set2Set operation to reduce the graph, followed by a 3-layer MLP. We use 64 hidden units in all of its parts, and LeakyReLU activations everywhere (except inside the GRU).

In the non-active-learning experiments, we train the proxy with a dataset of 300k molecules. To make the task interesting, we use 80% of molecules obtained from random trajectories, and 20% obtained from previous runs of RL agents. This extra 20% contains slightly higher scoring molecules that are

varied enough to allow for an interesting challenge. See Fig. 3 for the reward distribution. The exact dataset is provided on our github repository for reproducibility.

For the flow predictor $F$ we also use an MPNN, but it receives the block graph as input. This graph is a tree by construction. Each node in the graph is a learned embedding (each of the 105 blocks has its own embedding and each type of bond has an edge embedding). We again use a GRU over the convolution layer applied 10 times. For each stem of the graph (which represents an atom or block where the agent can attach a new block) we pass its corresponding embedding (the output of the 10 steps of graph convolution + GRU) into a 3-layer MLP to produce 105 logits representing the probability of attaching each block to this stem for MARS and PPO, or representing the flow $F(s, a)$ for GFlowNet; since each block can have multiple stems, this MLP also receives the underlying atom within the block to which the stem corresponds. For the stop action, we perform a global mean pooling followed by a 3-layer MLP that outputs 1 logit for each flow prediction. We use 256 hidden units everywhere as well as LeakyReLU activations.

For further stability we found that multiplying the loss for terminal transitions by a factor $\lambda_T > 1$ helped. Intuitively, doing so prioritizes correct predictions at the endpoints of the flow, which can then propagate through the rest of the network/state space via our bootstrapping objective. This is similar to using reward prediction as an auxiliary task in RL (Jaderberg et al., 2017).

Here is a summary of the flow model hyperparameters:

| | | |
|---|---|---|
| Learning rate | $5 \times 10^{-4}$ | |
| Minibatch size | 4 | # of trajectories per SGD step |
| Adam $\beta, \epsilon$ | $(0.9, 0.999), 10^{-8}$ | |
| # hidden & # embed | 256 | |
| # convolution steps | 10 | |
| Loss $\epsilon$ | $2.5 \times 10^{-5}$ | $\epsilon$ in (12) |
| Reward $T$ | 8 | |
| Reward $\beta$ | 10 | $\hat{R}(x) = (R(x)/T)^{\beta}$ |
| Random action probability | 0.05 | exploratory factor |
| $\lambda_T$ | 10 | leaf loss coefficient |
| $R_{min}$ | 0.01 | $R$ is clipped below $R_{min}$, i.e. $\hat{R}_{min} = (R_{min}/T)^{\beta}$ |

For MARS we use a learning rate of $2.5 \times 10^{-4}$ and for PPO, $1 \times 10^{-4}$. For PPO we use an entropy regularization coefficient of $10^{-6}$ and we set the reward $\beta$ to 4 (higher did not help). For MARS we use the same algorithmic hyperparameters as those found in Xie et al. (2021). For JT-VAE, we use the code provided by Jin et al. (2020) as-is, only replacing the reward signal with ours.

## A.5   Multi-Round Experiments

Algorithm 1 defines the procedure to train the policy $\pi_\theta$ and used in inner loop of the multi-round experiments in the hyper-grid and molecule domains. The effect of diverse generation becomes apparent in the multi-round setting. Since the proxy itself is trained based on the input samples proposed by the generative models (and scored by the oracle, e.g., using docking), if the generative model is not exploratory enough, the reward (defined by the proxy) would only give useful learning signals around the discovered modes. The oracle outcomes $O(x)$ are scaled to be positive, and a

hyper-parameter $\beta$ (a kind of inverse temperature) can be used to make the modes of the reward function more or less peaked.

---

**Algorithm 1:** Multi-Round Active Learning

---

**Input:** Initial dataset $D_0 = \{x_i, y_i\}, i = 1, \ldots, k$; $K$ for $TopK$ evaluation; number of rounds
$\qquad$ (outer loop iterations) $N$; inverse temperature $\beta$
**Result:** A set $TopK(D_N)$ of high valued $x$
**Initialization:**
Proxy $M$;
Generative policy $\pi_\theta$;
Oracle $O$;
$i = 1$;
**while** $i <= N$ **do**
$\qquad$ Fit $M$ on dataset $D_{i-1}$;
$\qquad$ Train $\pi_\theta$ with unnormalized probability function $r(x) = M(x)^\beta$ as target reward;
$\qquad$ Sample query batch $B = \{x_1, \ldots, x_b\}$ with $x_i \sim \pi_\theta$;
$\qquad$ Evaluate batch $B$ with $O$, $\hat{D}_i = \{(x_1, O(x_1)), \ldots, (x_b, O(x_b))\}$;
$\qquad$ Update dataset $D_i = \hat{D}_i \cup D_{i-1}$;
$\qquad$ $i = i + 1$;
**end**

---

### A.5.1 Hyper-grid

We use the Gaussian Process implementation from `botorch`[2] for the proxy. The query batch size of samples generated after each round is 16. The hyper-parameters for training the generative models are set to the best performing values from the single-round experiments.

The initial dataset only contains 4 of the modes. GFlowNet discovered 10 of the modes within 5 rounds, while MCMC discovered 10 within 10 rounds, whereas PPO managed to discover only 8 modes by the end (with $R_0 = 10^{-1}$).

### A.5.2 Molecules

The initial set $D_0$ of 2000 molecules is sampled randomly from the 300k dataset. At each round, for the MPNN proxy retraining, we use a fixed validation set for determining the stopping criterion. This validation set of 3000 examples is also sampled randomly from the 300k dataset. We use fewer iterations when fitting the generative model, and the rest of the hyper-parameters are the same as in the single round setting.

| method | Reward after 1800 docking evaluations | |
|---|---|---|
| | top-10 | top-100 |
| GFlowNet | $8.83 \pm 0.15$ | $7.76 \pm 0.11$ |
| MARS | $8.27 \pm 0.20$ | $7.08 \pm 0.13$ |

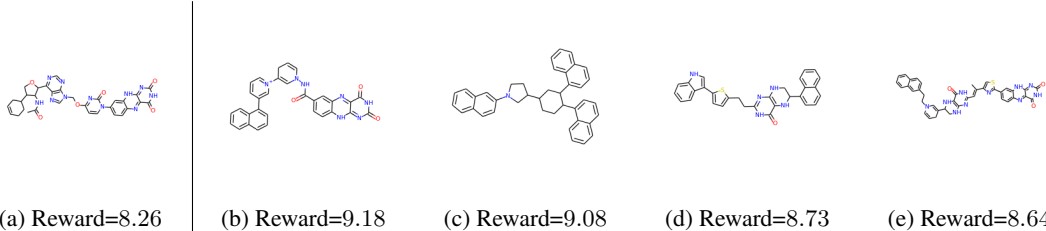

(a) Reward=8.26 $\qquad$ (b) Reward=9.18 $\qquad$ (c) Reward=9.08 $\qquad$ (d) Reward=8.73 $\qquad$ (e) Reward=8.64

Figure 9: (a) Highest reward molecule in $D_0$ in the multi-round molecule experiments. (b) Highest Reward molecule generated by GFlowNet. (c)-(e) Samples from the top-10 molecules generated by GFlowNet.

---

[2]http://botorch.org/

### A.6 Hypergrid Experiments

Let's first look at what is learned by GFlowNet. What is the distribution of flows learned? First, in Figure 10 (Left), we can observe that the distribution learned, $\pi_\theta(x)$, matches almost perfectly $p(x) \propto R(x)$ on a grid where $n = 2$, $H = 8$. In Figure 10 (Middle) we plot the visit distribution on all paths that lead to mode $s = (6, 6)$, starting at $s_0 = (0, 0)$. We see that it is fairly spread out, but not uniform: there seems to be some preference towards other corners, presumably due to early bias during learning as well as the position of the other modes. In Figure 10 (Right) we plot what the uniform distribution on paths from $(0, 0)$ to $(6, 6)$ would look like for reference. Note that our loss does not enforce any kind of distribution on flows, and a uniform flow is not necessarily desirable (investigating this could be interesting future work, perhaps some distributions of flows have better generalization properties).

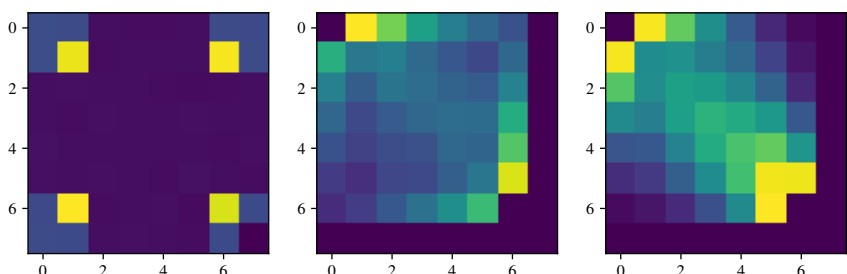

Figure 10: Grid with $n = 2$, $H = 8$. Left, the distribution $\pi_\theta(x)$ learned on the grid matches $p(x)$ almost perfectly; measured by sampling 30k points. Middle, the visit distribution on sampled paths leading to $(6, 6)$. Right, the uniform distribution on all paths leading to $(6, 6)$.

Note that we also ran Soft Actor Critic (Haarnoja et al., 2018) on this domain, but we were unable to find hyperparameters that pushed SAC to find all the modes for $n = 4$, $H = 8$; SAC would find at best 10 of the 16 modes even when strongly regularized (but not so much so that the policy trivially becomes the uniform policy). While we believe our implementation to be correct, we did not think it would be relevant to include these results in figures, as they are poor but not really surprising: as would be consistent with reward-maximization, SAC quickly finds a mode to latch onto, and concentrates all of its probability mass on that mode, which is the no-diversity failure mode of RL we are trying to avoid with GFlowNet.

Next let's look at the losses as a function of $R_0$, again in the $n = 4$, $H = 8$ setting. We separate the loss in two components, the leaf loss (loss for terminal transitions) and the inner flow loss (loss for non-terminals). In Figure 11 we see that as $R_0$ decreases, both inner flow and leaf losses get larger. This is reasonable for two reasons: first, for e.g. with $R_0 = 10^{-3}$, $\log 10^{-3}$ is a larger magnitude number which is harder for DNNs to accurately output, and second, the terminal states for which $\log 10^{-3}$ is the flow output are $100\times$ rarer than in the $R_0 = 10^{-1}$ case (because we are sampling states on-policy), thus a DNN is less inclined to correctly predict their value correctly. This incurs rare but large magnitude losses. Note that theses losses are nonetheless, small, in the order of $10^{-3}$ or less, and at this point the distribution is largely fit and the model is simply converging.

**GFlowNet as an offline off-policy method** To demonstrate this feature of GFlowNet, we train it on a fixed dataset of trajectories and observe what the learned distribution is. For this experiment we use $R(x) = 0.01 + \prod_i (\cos(50x_i) + 1) f_\mathcal{N}(5x_i)$, $f_\mathcal{N}$ is the normal p.d.f., $n = 2$ and $H = 30$. We show results for two random datasets. First, in Figure 12 we show what is learned when the dataset is sampled from a uniform random policy, and second in Figure 13 when the dataset is created by sampling points uniformly on the grid and walking backwards to the root to generate trajectories. The first setting should be much harder than the second, and indeed the learned distribution matches $p(x)$ much better when the dataset points are more uniform. Note that in both cases many points are left out intentionally as a generalization test.

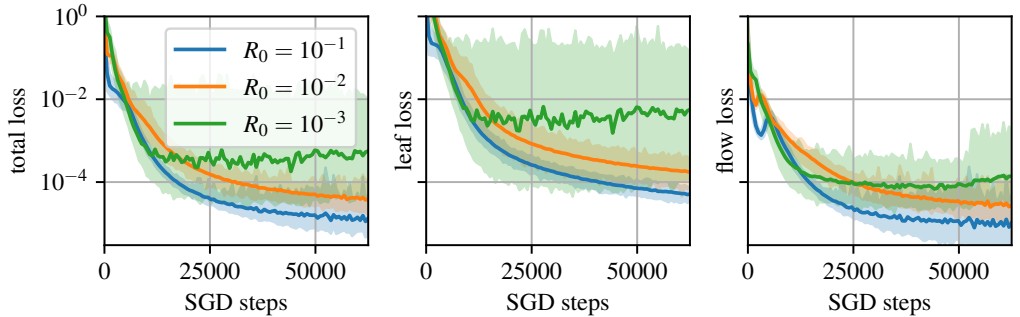

Figure 11: Losses during training for the "corners" reward function in the hypergrid, with $n = 4$, $H = 8$. Shaded regions are the min-max bounds.

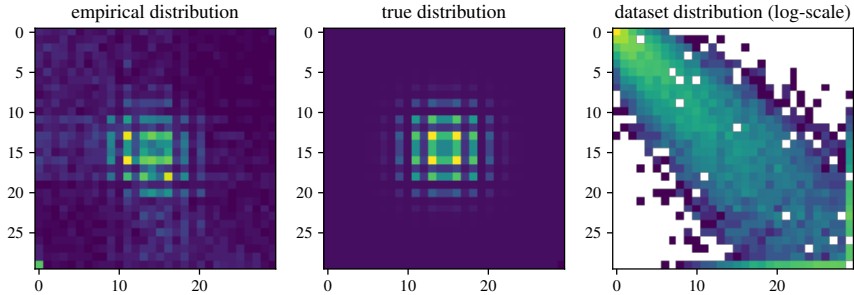

Figure 12: Grid with $n = 2$, $H = 30$. Left, the learned distribution $\pi_\theta(x)$. Middle, the true distribution. Right, the dataset distribution, here generated by executing a uniform random policy from $s_0$.

These results suggest that GFlowNet can easily be applied offline and off-policy. Note that we did not do hyperparameter search on these two plots, these are purely illustrative and we believe it is likely that better generalization can be achieved by tweaking hyperparameters.

### A.7 GFlowNet results on the molecule domain

Here we present additional results to give insights on what is learned by our method, GFlowNet.

Let's first examine the numerical results of Figure 4:

| method | Reward at $10^5$ samples | | |
|---|---|---|---|
| | top-10 | top-100 | top-1000 |
| GFlowNet | $8.36 \pm 0.01$ | $8.21 \pm 0.03$ | $7.98 \pm 0.04$ |
| MARS | $8.05 \pm 0.12$ | $7.71 \pm 0.09$ | $7.13 \pm 0.19$ |
| PPO | $8.06 \pm 0.26$ | $7.87 \pm 0.29$ | $7.52 \pm 0.26$ |
| | Reward at $10^6$ samples | | |
| GFlowNet | $8.45 \pm 0.03$ | $8.34 \pm 0.02$ | $8.17 \pm 0.02$ |
| MARS | $8.31 \pm 0.03$ | $8.03 \pm 0.08$ | $7.64 \pm 0.16$ |
| PPO | $8.25 \pm 0.12$ | $8.08 \pm 0.12$ | $7.82 \pm 0.16$ |
| | Reward for $10^6$-equivalent compute | | |
| JT-VAE + BO | 6.03 | 5.86 | 5.31 |

These are means and standard deviations computed over 3 runs. We see that GFlowNet produces significantly better molecules. It also produces much more diverse ones: GFlowNet has a mean

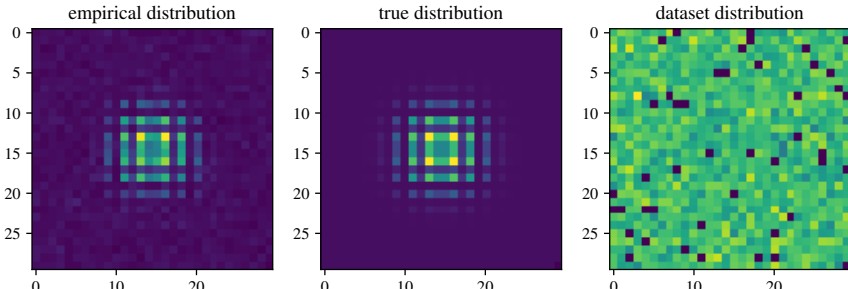

Figure 13: Grid with $n = 2$, $H = 30$. Left, the learned distribution $\pi_\theta(x)$. Middle, the true distribution. Right, the dataset distribution, here generated by sampling a point uniformly on the grid and sampling random parents until $s_0$ is reached, thus generating a training trajectory in reverse.

pairwise Tanimoto similarity for its top-1000 molecules of $0.44 \pm 0.01$, PPO, $0.62 \pm 0.03$, and MARS, $0.59 \pm 0.02$ (mean and std over 3 runs). A random agent for this environment would yield an average pairwise similarity of $0.231$ (and very poor rewards).

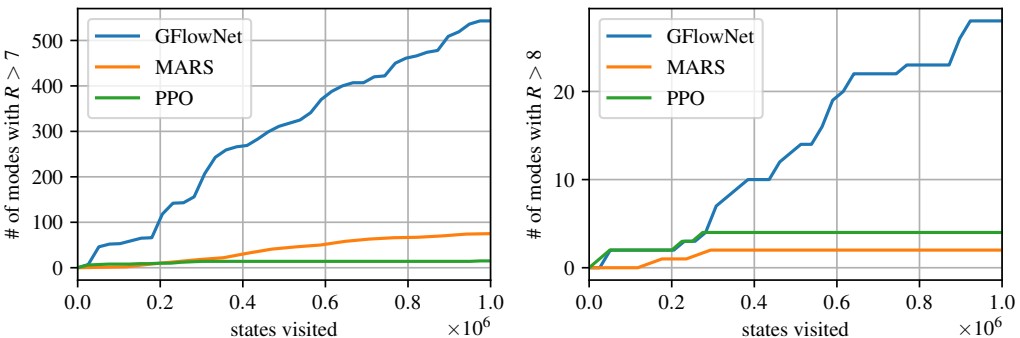

Figure 14: Number of Tanimoto-separated modes found above reward threshold $T$ as a function of the number of molecules seen. See main text. Left, $T = 7$. Right, $T = 8$.

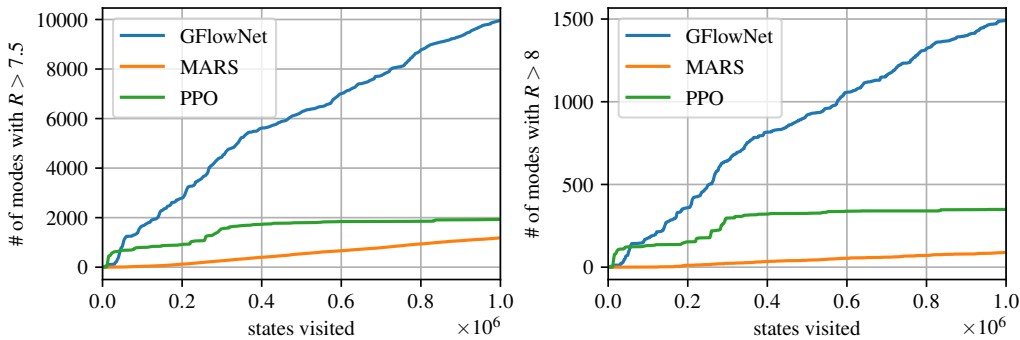

Figure 15: Number of diverse Bemis-Murcko scaffolds (Bemis and Murcko, 1996) found above reward threshold $T$ as a function of the number of molecules seen. Left, $T = 7.5$. Right, $T = 8$.

We also see that GFlowNet produces much more diverse molecules by approximately counting the number of modes found within the high-reward molecules. Here, we define "modes" as molecules with an energy above some threshold $T$, at most similar to each other in Tanimoto space at threshold $S$. In other words, we consider having found a new mode representative when a new molecule has a

Tanimoto similarity smaller than $S$ to every previously found mode's representative molecule. We choose a Tanimoto similarity $S$ of 0.7 as a threshold, as it is commonly used in medicinal chemistry to find similar molecules, and a reward threshold of 7 or 8. We plot the results in Figure 14. We see that for $R > 7$, GFlowNet discovers many more modes than MARS or PPO, over 500, whereas MARS only discovers less than 100.

Another way to approximate the number of modes is to count the number of diverse Bemis-Murcko scaffolds present within molecules above a certain reward threshold. We plot these counts in Figure 5, where we again see that GFlowNet finds a greater number of modes.

Next, let's try to understand what is learned by GFlowNet. In a large scale domain without access to $p(x)$, it is non-trivial to demonstrate that $\pi_\theta(x)$ matches the desired distribution $p(x) \propto R(x)$. This is due to the many-paths problem: to compute the true $p_\theta(x)$ we would need to sum the $p_\theta(\tau)$ of all the trajectories that lead to $x$, of which there can be an extremely large number. Instead, we show various measures that suggest that the learned distribution is consistent with the hypothesis the $\pi_\theta(x)$ matches $p(x) \propto R(x)^\beta$ well enough.

In Figure 16 we show how $F_\theta$ partially learns to match $R(x)$. In particular we plot the inflow of leaves (i.e. for leaves $s'$ the $\sum_{s,a:T(s,a)=s'} F(s,a)$) as versus the target score ($R(x)^\beta$).

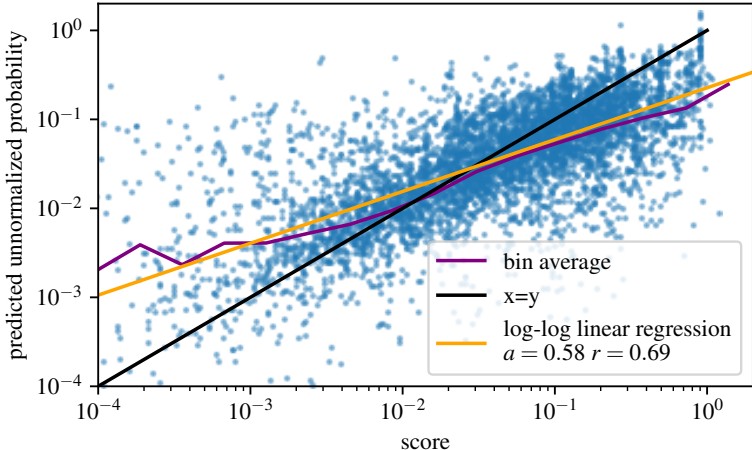

Figure 16: Scatter of the score ($R(x)^\beta$) vs the inflow of leaves (the predicted unnormalized probability). The two should match. We see that a log-log linear regression has a slope of $0.58$ and a $r$ of $0.69$. The slope being less than 1 suggests that GFlowNet tends to underestimate high rewards (this is plausible since high rewards are visited less often due to their rarity), but nonetheless reasonably fits its data. Here $\beta = 10$. We plot here the last 5k molecules generated by a run.

Another way to view that the learned probabilities are self-consistent is that the histograms of $R(x)/Z$ and $\hat{p}_\theta(x)/Z$ match, where we use the predicted $Z = \sum_{a \in \mathcal{A}(s_0)} F(s_0, a)$, and $\hat{p}_\theta(x)$ is the inflow of the leaf $x$ as above. We show this in Figure 17.

In terms of loss, it is interesting that our models behaves similarly to value prediction in deep RL, in the sense that the value loss never goes to 0. This is somewhat expected due to bootstrapping, and the size of the state space. Indeed, in our hypergrid experiments the loss does go to 0 as the model converges. We plot the loss separately for leaf transitions (where the inflow is trained to match the reward) and inner flow transitions (at visited states, where the inflow is trained to match the outflow) in Figure 18.

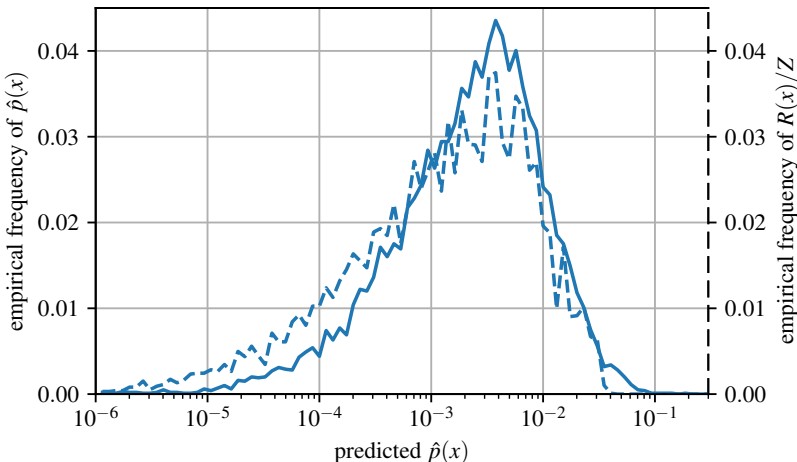

Figure 17: Histogram of the predicted density vs histogram of reward. The two should match. We compute these with the last 10k molecules generated by a run. This plot again suggests that the model is underfitted. It thinks the low-reward molecules are less likely than they actually are, or vice-versa that the low-reward molecules are better than they actually are. This is consistent with the previous plot showing a lower-than-1 slope.

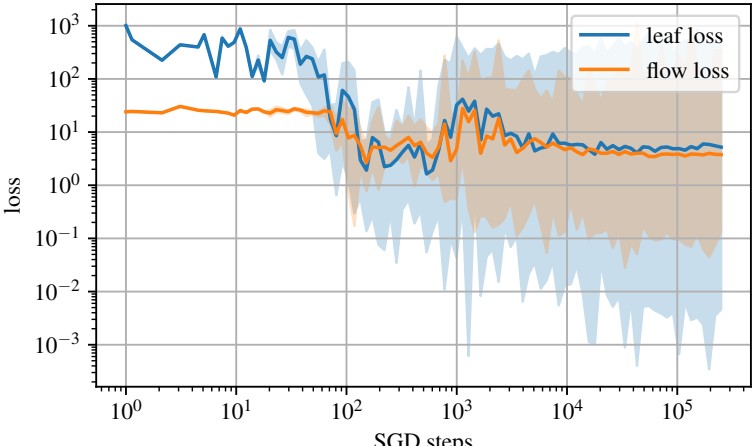

Figure 18: Loss as a function of training for a typical run of GFlowNet on the molecule domain. The shaded regions represent the min-max over the interval. We note several phases: In the initial phase the scale of the predictions are off and the leaf loss is very high. As prediction scales adjust we observe the second phase where the flow becomes consistent and we observe a dip in the loss. Then, as the model starts discovering more interesting samples, the loss goes up, and then down as it starts to correctly fit the flow over a large variety of samples. The lack of convergence is expected due to the massive state space; this is akin to value-based methods in deep RL on domains such as Atari.