# OpenReview forum: "Flow Network based Generative Models for Non-Iterative Diverse Candidate Generation"
_NeurIPS.cc/2021/Conference — NeurIPS 2021 Poster_

### Official Review · Reviewer_3Yk2 · 2021-07-11

**Rating:** 7
**Confidence:** 3

**Summary:**

The paper considers generative processes for which state transitions form a DAG. The authors view such a generative process as a flow in a graph—with a single source and sinks connected to final objects. The authors set objects' probabilities being proportional to the reward function and train a generative model by enforcing flow-matching conditions. The main contribution of the paper is a new generative approach for unnormalized distributions.

**Limitations And Societal Impact:**

The authors adequately addressed the limitations.

**Main Review:**

The proposed approach seems novel and is well described in the paper. The authors provide theoretical and practical details, as well as intuition behind the method. Experimental evaluation is sufficient, and results are demonstrated on a toy problem as well as on a practical task of molecular property optimization. Provided results in supplementary (Fig. 13 and Fig. 14) demonstrate huge advantage of the proposed approach.

Some comments are provided below:
1. More details on relation of the proposed approach and Bayesian networks would be profitable. Currently it seems as if model definition using a Bayesian network would yield a similar approach (e.g., flow-matching condition will be the same as total probability formulta).
2. Experimental results should be provided on standard datasets. For example, optimization of logP and QED as provided in multiple papers, including GrammarVAE (https://arxiv.org/pdf/1703.01925.pdf). Such experiments are necessary to confirm that the proposed approach indeed yields good results against strong baselines.
3. Diverse generation as mentioned in the title of the paper, is not built in the model and is a side-effect of the approach. I would recommend conducting experiments on the objective functions with high mode imbalance (e.g., 10k molecules with high reward in one mode and 5 molecules with high reward in another).

Update: The authors answered my questions, I keep my original score.

**Time Spent Reviewing:**

4

---

> ### Author Response · Authors · 2021-08-10
> **Answers to questions**
>
>
> Thank you for your review, we will incorporate these comments to our paper.
>
> > More details on relation of the proposed approach and Bayesian networks would be profitable. Currently it seems as if model definition using a Bayesian network would yield a similar approach (e.g., flow-matching condition will be the same as total probability formulta).
>
> We had not thought about this analogy, but there certainly are some conceptual links to be made. The nodes in a GFlowNet have a more specialized semantics than those in a Bayes Net, but a GFlowNet can be seen as a special kind of BayesNet, with its DAG structure and random variables at nodes and conditional probabilities linking them (however the parametrization is very different, not based on the conditional of a variable given its parents, but the conditional probability of mutually exclusive children being true given some parent). A GFlowNet node $s$ represents the event "the outcome $X$ was generated by first constructing $s$", where $X$ is a sample from the GFlowNet (e.g. a complete molecule) while $s$ can be on the path (the sequence of transformations) which goes from the empty initial state to the construction of $X$. There is an order relation "$<$" that relates successive states, i.e., $s_t<s_{t+1}$. These special properties of GFlowNets enable particular algorithms which are not applicable with BayesNets in general.
>
>
> > Experimental results should be provided on standard datasets. For example, optimization of logP and QED as provided in multiple papers, including GrammarVAE (https://arxiv.org/pdf/1703.01925.pdf). Such experiments are necessary to confirm that the proposed approach indeed yields good results against strong baselines.
>
> We did not optimize for logP and QED because on their own these are fairly easy to maximize.
> Just to be sure, we ran GFlowNet on logP and QED. For logP we quickly find molecules with a >20 logP, which at this point is biologically uninteresting (for reference, ibuprofen's logP is between 3.5 and 4). For QED, we also quickly find molecules with the maximum possible QED in our action space, which is 0.948 (in fact our top-1000 is >0.94 after 100k molecules seen). We will add those results to the appendix for completeness.
>
> > Diverse generation as mentioned in the title of the paper, is not built in the model and is a side-effect of the approach. I would recommend conducting experiments on the objective functions with high mode imbalance (e.g., 10k molecules with high reward in one mode and 5 molecules with high reward in another).
>
> We are confident that GFlowNet does diverse generation (see figures 2, 13 and 14), but it would indeed be interesting to see how robust it is to bad reward landscapes. We will try to include such an experiment (and hopefully we can report it before the end of the discussion period).
> It might interesting to note here that the molecule landscape is already a bit like this, although not as imbalanced as you suggest testing. For some good molecules, removing or adding an atom can have a big effect and make them terrible molecules, whereas other molecules are robust to being "edited". These naturally make for peaky and wide modes respectively.

---

### Official Review · Reviewer_UVLM · 2021-07-16

**Rating:** 6
**Confidence:** 1

**Summary:**

The paper tries to tackle a scenario where the goal is not to learn to generate a sequence of actions that has the highest return but rather to generate a diverse set of trajectories with high returns. They do so by using the given positive return function (which is given by the oracle function) to formulate a generative policy that samples with a probability proportional to the return.  They used a flow network to express the MDP and its value function where the value of a state and action pairs is defined by the flow between the two nodes in the flow network, and finally, the flow consistency equations are used as a learning objective.

**Limitations And Societal Impact:**

The paper is a bit hard to follow.

**Main Review:**

The idea of the paper is very interesting but the paper is hard to follow as the paper borrows notations and concepts from RL but not in the exact same setting so it makes it a bit confusing for the readers. I personally feel like the paper could be made more intuitive and much easy to follow if it uses more direct and intuitive concepts and pushes less hard to connect with concepts in RL.

There are some minor comments and questions:
Comments:
Line 71: The RL “agent” should have large support of what?
Line 84: Missing citations of PPO and MCMC
Figure 3: MARS looks to approximate better the proxy dataset distribution.
Figure3: Didn’t understand the shifting of the density to the right.



**Time Spent Reviewing:**

8 hours

---

> ### Author Response · Authors · 2021-08-10
> **Answers to questions**
>
>
> Thank you for your review, we will adjust the paper to be more explicit about notation.
>
> > the paper could be made more intuitive and much easy to follow if it uses more direct and intuitive concepts and pushes less hard to connect with concepts in RL.
>
> This is a very good point, and we are indeed working on a formalization which avoids borrowing the usual semantics of RL, to minimize confusion. Thanks for the suggestion.
>
> > Line 71: The RL “agent” should have large support of what?
>
> Here we mean support over the state space, i.e. have a non-zero probability of reaching states. We will clarify the text.
> In terms of off-policy methods, one way to think about it is that if we are estimating a quantity depending on policy $\pi$ (e.g. $V^\pi$) while running policy $\mu$, then for any state which $\pi$ would visit, $\mu$ ideally needs to have a non-zero probability of visiting it as well (but these probabilities do not need to be equal). If an RL method is truly off-policy, this should be enough to learn $V^\pi$, even if $\pi\neq\mu$.
>
> > Figure 3: MARS looks to approximate better the proxy dataset distribution
>
> The dataset distribution and MARS's (or GFlowNet's) distribution should not match. For a uniform dataset, we should have that $p(x) = 1/|\mathcal{X}|$, but MARS and GFlowNet should be so that $p(x) = R(x) / Z$ (see Eq. (1)).
>
> As such, these will not be the same. In fact, the probability of $x$s with a higher $R(x)$ should be increased (intuitively, since it becomes $\propto R(x)$ rather than $\propto c$, one can imagine that if $R(x) > c$ then $p(x)$ should be larger in the second distribution). This explains why in Figure 3, the proxy dataset distribution is more to the left than the other distributions.
>
> > Figure3: Didn’t understand the shifting of the density to the right.
>
> We compare sampling from $p(x) \propto R(x)^1$ with sampling from $p(x) \propto R(x)^4$ (raising to power 4). In the second distribution, $x$s with higher $R$s should have more probability. In Figure 3 we show the histogram of $R$, thus if the higher $R$s have more probability, the distribution curve will be higher on the right hand side (since the x-axis goes from 0 to 9). This will visually look like the densities are shifted to the right.
>
> Just to be clear, here's an example. If $\mathcal{X} = \{x_1, x_2\}$, and $R(x_1)=1,R(x_2)=2$. Then for $p(x) \propto R(x)^1$, $p(x_1)=1/(1+2)=1/3$ and $p(x_2)=2/3$. For $p(x) \propto R(x)^4$, $p(x_1) = 1/(1^4+2^4) = 1/17$, and $p(x_2)=16/17$. $x_2$ becomes much more likely because it has a higher reward and we are using $\beta=4$.

---

> > ### Comment · Reviewer_UVLM · 2021-08-17
> > **Commenting on authors' rebuttal**
> >
> > Thank you for the clarification. It clarified most of my concerns. After reading the comments of reviewer 1AuD and others,  I have a better understanding of the paper.  The idea is interesting and new but I still believe that the paper is relatively confusing for the common readers. Therefore I decided to keep my original score.

---

### Official Review · Reviewer_1AuD · 2021-07-16

**Rating:** 7
**Confidence:** 3

**Summary:**

This paper proposes an approach (called GFlowNet) that uses ideas from reinforcement learning to learn a "policy" for sampling from a discrete distribution over a set of objects ("terminal states"), where these objects can be traversed between using a sequence of "actions". Specifically, the authors consider distributions proportional to some (non-negative) reward function, such that one is more likely to sample better performing points (although I believe the approach is general enough to sample from any unnormalized distribution).

In the experimental section the authors first consider a toy problem to highlight the advantages of their approach before moving on to consider the problem of designing new molecules, where the "state" is the current molecule, and the "actions" are defined using a vocabulary of fragments which one can add to the current state (as well as the stop action).  Here, the authors show GFlowNet is able to find  a set of better and more diverse molecules more quickly than previous MCMC and PPO approaches.


**More specific details about GFlowNet:**
In GFlowNet, states and actions can be described using a flow network (i.e. a transportation network). Here the states are represented using nodes and the actions as edges (the edge weight, ie flow, describes the probability of taking an action). The authors always consider starting from the same state and actions cannot return to previously seen states (ie fragments can be added to molecules but not taken away). Moreover, one can arrive at the same state with different actions (i.e. different fragments could be used to create the same molecule). Overall, this means that the flow network can be represented using a DAG (directed acyclic graph, see Figure 1).

The core idea of GFlowNet is then about how to learn the policy over the actions (ie the flows) such that when sampling actions from this policy, you sample from a predefined distribution over terminal states. To this end, the authors derive a flow matching objective (equation 9 -- the flow incoming to a state must match the outgoing flow) which can be differentiated through to learn the parameters parameterizing the policy. The authors prove that this loss can be used off-policy.


**Ethical Concerns:**

None.

**Limitations And Societal Impact:**

# 5 Limitations

a. The authors briefly talk about the limitations of the approach in section 5. The main limitation they draw attention to is the challenge of moving closer to the local maxima of the reward function in the latter stages of optimization. To resolve this they discuss combining their method with local optimization techniques; however, I wonder whether the temperature approach they discuss in the earlier part of their paper (combined with some annealing scheme) could also be used here?

b. One limitation the authors do not mention is how the method scales in terms of the size of the state and action space. The loss function requires for every current state the sum over all previous states and actions that may have led to the current state (see term 1 of Eq.9). I assume this may become intractable for very large state-action spaces (and the flows one is trying to model become very small). Can one approximate the sum using a subset? Also what about continuous state/action spaces?




# 6 Societal impact
The authors state that they foresee no negative social impacts of their work (line 379). While I do not believe this work has the potential for significant negative social impact (and I'm not quite sure if/how I'm meant to review this aspect of their work), the authors could always mention the social impact of increased automation, or the risks from the dual use of their method, etc.


**Main Review:**

# 1 Summary of My Review
I found the approach presented in this paper very interesting and it appears to achieve impressive results. However, on a more negative note, some of the paper was hard to follow and the clarity could be improved in places (see main review below). I believe this would be easy enough to fix, and therefore, I have gone for a higher "overall score" and think this paper should be accepted. I have gone with a lower confidence score as I'm less knowledgable about how this idea relates to recent work in RL.

# 2 Main Review
## 2.1 Originality
In the molecule domain, the idea of using RL-based ideas to sample from the distribution of all molecules (weighted by their scores), rather than just find the top molecule, seems novel. Obtaining a set of diverse **and** well-performing molecules is an important problem and the proposed approach is an interesting way to do so.

Unfortunately, I am less familiar with the recent, more general RL literature in this domain so feel less qualified to discuss the paper's originality in this respect, but the approach is fairly general and could hopefully also be of interest here.

The idea of visited states being equivalent (and so the state-action space being best represented with a DAG rather than a tree) has been considered in the context of Monte Carlo Tree Search, which the authors might briefly want to mention, e.g see the discussion about "transpositions" in Section 5.2.4 of:
> Browne, C. B., Powley, E., Whitehouse, D., Lucas, S. M., Cowling, P. I., Rohlfshagen, P., Tavener, S., Perez, D., Samothrakis, S. and Colton, S. (2012) ‘A Survey of Monte Carlo Tree Search Methods’, IEEE Transactions on Computational Intelligence in AI and Games, 4(1), pp. 1–43.

## 2.2 Quality
a. As far as I can tell the paper is technically sound (although I did not look through the proofs in the appendix). The approach to make the method computationally tractable (leading to Eq 9) seems reasonable although I would have been interested to hear how dependent the results presented later are on the value of $\epsilon$ chosen.

b. One other currently popular way to sample from the distribution over molecules is using autoencoders with autoregressive decoders, for instance the junction tree VAE of Jin et al. (2020) for which the fragment-based scheme in this paper is said to be based upon. This seems like an obvious baseline to compare to in the experiments but is omitted.

c. I found some of the experimental evaluation for the molecule task a little confusing: the authors invent a method for sampling from a distribution, yet evaluate it mostly in terms of how well it discovers the modes of the distribution (rather than how well it samples from the distribution as a whole). I realize that this is useful when the distribution is defined by a reward function, but what about when sampling more general distributions? (related to question 3.a. below)

## 2.3 Clarity

a. One area for improvement is perhaps the paper's clarity. For instance, section 2 is rather dense and I had to read it twice to understand what was going on. Perhaps adding more signposting to the beginning of this section would allow the reader to follow its various parts.

I also got a bit confused by the notation in this section. For instance $\tau$ suddenly appears in section 2.2 to denote the trajectory, whereas earlier on in this section I believe $\vec{a}$ was used instead (eg line 123). (Or do these variables mean different things....?)

Likewise, I got a bit confused with Equation 10. I think the second part of this should be $L_{\theta^\ast}(\tau) = 0 \quad \forall \tau \sim P(\tau)$, i.e. with $\tau$ in place of $\theta$? (On a more minor note I was also uncertain as to why $L_{\theta^\ast}(\tau) $ had to equal zero, the approach can also deal with loss functions with any minimum right?)

b. In section 4.1 (ie the toy experiment on the grid domain) I didn't understand why the states $s$ could not live in the same space as $x$ (defining the reward)? To convert from $s$ to $x$ you had to add the vector $-\mathbf{1}$. This just seemed to add unnecessary complexity to the setup and made it harder to understand?

c. I've only briefly glanced over the code, which hopefully should aid reproducibility. One thing that could be added here though is a description of which scripts to run (including runtime arguments) to actually reproduce the experiments.


## 2.4 Significance
This paper presents interesting theoretical results as well as strong empirical performance. In the molecule optimization task it is nice to see the focus on obtaining a _diverse set_ of top molecules rather than just the top molecule score (although it would be nice if the top-1 score was also reported somewhere for completeness). The authors consider more interesting (and difficult) molecular optimization tasks (based on docking) than much of the previous work, and it is exciting to see improvements wrt recent (and reasonable) baselines.

I think the ideas that GFlowNet is built around are interesting, particularly of how to sample from a distribution of discrete objects when the transition function (i.e. "action" using the paper's terminology) is irreversible. Hopefully this will inspire further work in this direction.


# 3 Clarifying Questions

a. My understanding is that (if trained/run for long enough) the MCMC baseline and GFlowNet should ultimately sample from the same distribution, i.e. that defined by:
$\pi(x) \propto R(x)$. However, in figure 3 it appears that GFlowNet's distribution is different (the mode is higher and the distribution has wider tails). Does this just mean the MCMC method has not converged? Is there any way to compare against the "true" distribution (for instance running the MCMC method for much longer, or using very many random trajectories and importance sampling)?

b I am currently having a bit of trouble understanding figure 4. On lines 330-336 the authors state that when sampling 300k molecules using random trajectories that they obtain over 100 molecules with scores above 8 (233 such molecules to be precise). Looking at figure 4 the top-100 average for MARS is below 8 after visiting 300k molecules. Does this mean that it is worse than random search? Is this expected?

c For the MCMC methods (e.g. MARS) did the authors consider thinning?

d A related question. For GFlowNet I assume every state is valid? If so I was wondering if the authors ever considered counting the reward for every state visited by GFlowNet rather than just the terminal ones. In some ways can one think of the "stop action" as a learnable thinning function?

e I apologize if I missed this but what is the initial state for the molecule tasks?


# 4 Very Minor Comments

i. Lines 105, 138 and elsewhere on page 3 the word "surjective" is used. I think "non-injective" is meant here instead...?

ii. I found the figure on the right below line 265 helpful for understanding the toy problem (so thank you to the authors for this). However, its axis is too small and unreadable -- this should be made larger. Similarly, Figure 7 in the appendix could be bigger (especially given that there are no space constraints in that section).

iii. There are some weird black horizontal-ish lines on Figure 3 (at least on the two pdf viewers that I tried). Did these mean anything or were they just some left over plotting artifacts?


**Time Spent Reviewing:**

6

---

> ### Author Response · Authors · 2021-08-10
> **Answers to questions**
>
>
> Thank you for your detailed review!
>
> > how dependent the results presented later are on the value of $\epsilon$ chosen
>
> It is true that this hyperparameter is worth discussing more. We will refine our explanation.
> Ideally, $\epsilon$ should be set to be about as low as the minimum reward one cares about, or close to the minimum useful reward of the environment. This would help the model spend less capacity on the states with flow lower than that. Empirically we have found that this choice was stable, but did not conduct an extensive investigation.
>
> > One other currently popular way to sample from the distribution over molecules is using autoencoders with autoregressive decoders
>
> This is a good point, and we shall add such comparisons. The reason we did not explore in that direction is that in our active learning setting we are not given a set of 'good examples' (the usual approach with generative models like VAEs, GANs, etc): instead we are given a goodness function (the reward), which includes information about 'bad examples' and all the shades in between good and bad. It would seem wasteful to throw away most of the information acquired acquired through active learning and only keep the tail of the distribution of collected experimental results.
>
> > section 2 is rather dense and I had to read it twice to understand what was going on
>
> We will try to clarify and be more explicit about the function of each part.
>
> > ($L_{\theta^*}(\theta)$)... should be $L_{\theta^*}(\tau)$
>
> You are right. Thanks for catching this typo. You are also right that in principle we can use any loss function but the ones we defined have value 0 when the flows are matched, which yields the convenient outcome of this theorem (that the global minimizer in function space are guaranteed to match the flows.)
>
> > To convert from $s$ to $x$ you had to add the vector -1
>
> When we constructed this environment, we wrote a reward for $[-1,1]^n$, and wanted to subdivide an overlapping grid in $H$ parts and vary $H$, thus our presentation; but you are correct that this distinction is unnecessary. We will simplify this passage.
>
>
> > My understanding is that (if trained/run for long enough) the MCMC baseline and GFlowNet should ultimately sample from the same distribution, i.e. that defined by: \pi(x) \propto \R(x). However, in figure 3 it appears that GFlowNet's distribution is different (the mode is higher and the distribution has wider tails). Does this just mean the MCMC method has not converged? Is there any way to compare against the "true" distribution (for instance running the MCMC method for much longer, or using very many random trajectories and importance sampling)?
>
> This is correct, MCMC and GFlowNet should sample from the same distribution. Here we suspect that MCMC is missing high-reward modes, thus the mismatch in distributions.
> In general in molecule space it would take orders of magnitude more compute to converge to (and compute) the "true" distribution, which we unfortunately could not afford. In the hypergrid setting, we verify that both methods converge to the true distribution.
>
>
> > I am currently having a bit of trouble understanding figure 4. On lines 330-336 the authors state that when sampling 300k molecules using random trajectories that they obtain over 100 molecules with scores above 8 (233 such molecules to be precise). Looking at figure 4 the top-100 average for MARS is below 8 after visiting 300k molecules. Does this mean that it is worse than random search? Is this expected?
>
> Thanks for this observation, we went back and checked how the initial dataset we're using in the paper has been generated, and it is indeed not entirely random data: it includes a bit of data generated by past RL agents that we have run on this problem (the methods in this paper are used within a larger scope drug-discovery project). We will edit the paper to reflect this.
> Note that we provide this dataset in the supplementary material, so the exact setup presented in the paper is reproducible.
>
> >  For the MCMC methods (e.g. MARS) did the authors consider thinning? For GFlowNet I assume every state is valid? If so I was wondering if the authors ever considered counting the reward for every state visited by GFlowNet rather than just the terminal ones. In some ways can one think of the "stop action" as a learnable thinning function?
>
> We have not used thinning in MCMC methods; for finding the top-k this would not be beneficial: if the chain did not visit enough modes, then subsampling it would not help with getting samples from those unvisited modes. For GFlowNet, your suggestion is a good one which we actually have started implementing since submitting the paper, since every state is valid we can indeed consider a virtual stop action after every state to be used as a learning signal.
>
> On the analogy between the stop action and thinning: thinning helps to induce iid-ness in a Markov Chain because samples close to each other in a chain are correlated. In GFlowNet, every sample is generated as an independent episode, which the stop action ends. Thus the stop action does not break any correlations. However, you are right that some of the intermediate states could be valid solutions and we could use something inspired like thinning (but no need to subsample) and consider all the subsequences of each terminated sequence as a possible outcome. However, if not done properly, this could break the main property of having sampling probabilities matching the reward function.
>
>
> > I apologize if I missed this but what is the initial state for the molecule tasks?
>
> The initial state is the "empty molecule", to which any block can be attached. We will make this clearer in the text.
>
>
> > the word "surjective" is used. I think "non-injective" is meant here instead.
>
> Indeed, the emphasis in lines 105 and 138 is to highlight the case when the bijective approach breaks down, so the sentence should say non-injective. However, the mathematical framework we designed includes the bijective case as a special case, which is why we used *surjective* everywhere. But you're right that in these sentences it would be more correct as you pointed out to say *non-injective*. We will fix that.
>
> > The authors briefly talk about the limitations of the approach in section 5. The main limitation they draw attention to is the challenge of moving closer to the local maxima of the reward function in the latter stages of optimization. To resolve this they discuss combining their method with local optimization techniques; however, I wonder whether the temperature approach they discuss in the earlier part of their paper (combined with some annealing scheme) could also be used here?
>
> This is correct, using a higher temperature is a valid approach. The reasoning for having a separate local optimization is that these can be run in parallel, without affecting GFlowNet's exploration and learning procedure.
>
> > One limitation the authors do not mention is how the method scales in terms of the size of the state and action space. The loss function requires for every current state the sum over all previous states and actions that may have led to the current state (see term 1 of Eq.9). I assume this may become intractable for very large state-action spaces (and the flows one is trying to model become very small). Can one approximate the sum using a subset? Also what about continuous state/action spaces?
>
> This is indeed a limitation, if a state has many parents then it becomes more expensive to compute the loss function. In the case of molecules, this is small enough, but it could become a problem for larger objects. Since submitting the paper, we have found alternative formulations to compute the flow that do not require this sum (and that possibly work in continuous action spaces), but these are different enough that they would warrant a follow-up paper.

---

> > ### Comment · Reviewer_1AuD · 2021-08-16
> > **Commenting on authors' rebuttal/other reviews**
> >
> > I thank the authors for their rebuttal. I had a few comments/questions:
> >
> > ### 1. The "random" molecule dataset (replying to 3b)
> > > "Thanks for this observation, we went back and checked how the initial dataset we're using in the paper has been generated, and it is indeed not entirely random data: it includes a bit of data generated by past RL agents that we have run on this problem (the methods in this paper are used within a larger scope drug-discovery project). We will edit the paper to reflect this. Note that we provide this dataset in the supplementary material, so the exact setup presented in the paper is reproducible."
> >
> > This seemed rather strange: did you mean to use this dataset? Just so I understand correctly: this "random"dataset was only used for training the scoring proxy, not the generative models? Also, it would be interesting to hear more about these past RL agents, are they better than PPO?
> >
> >
> > ### 2. Comparison to standard baselines (replying to 2.2b)
> > > "This is a good point, and we shall add such comparisons. The reason we did not explore in that direction is that in our active learning setting we are not given a set of 'good examples' (the usual approach with generative models like VAEs, GANs, etc): instead we are given a goodness function (the reward), which includes information about 'bad examples' and all the shades in between good and bad. It would seem wasteful to throw away most of the information acquired acquired through active learning and only keep the tail of the distribution of collected experimental results."
> >
> > I'm not completely sure I follow the argument here. The junction tree VAE I mention in section 2.2 would also see the continuous "goodness function" in its BayesOpt routine and so should also obtain information from the bad examples. Anyway, I'm glad to hear that you are adding comparisons to such approaches (which seems maybe also of interest to Reviewer 3Yk2)!
> >
> >
> > ### 3. Scaling to larger action spaces (replying to 5b)
> > > "This is indeed a limitation, if a state has many parents then it becomes more expensive to compute the loss function. In the case of molecules, this is small enough, but it could become a problem for larger objects. Since submitting the paper, we have found alternative formulations to compute the flow that do not require this sum (and that possibly work in continuous action spaces), but these are different enough that they would warrant a follow-up paper."
> >
> > This seems very interesting and reasonable to postpone to future work. I look forward to reading!

---

> > > ### Author Response · Authors · 2021-08-16
> > > **Additional answers**
> > >
> > > > did you mean to use this dataset?
> > >
> > > Yes, its erroneous description in the paper probably stemmed from our focusing our time on explaining GFlowNet. The dataset is ~80% generated from random walks, the rest is from older iterations of actor-critic and PPO agents (before we invented GFlowNet, PPO was our best method for molecule generation) trained on docking scores. The reasoning for this mix was that populating the dataset with a few more high-scoring molecules (according to the docking oracle) would make for a more interesting generative task.
> > >
> > > The end goal of this project is to do active learning (i.e. query a docking oracle, and eventually even more powerful/precise oracles) with batches from our generative models, and so incrementally build a better and better dataset (and by extension, proxy). Note that the results of Section 4.3 & Figure 6 demonstrate this capability fairly well, starting from only 2000 molecules and interacting directly with the docking oracle.
> > >
> > > > Just so I understand correctly: this "random" dataset was only used for training the scoring proxy, not the generative models?
> > >
> > > This is correct. None of the generative models have access to the dataset that the proxy is trained with. They have to discover the high-reward parts of the state space on their own.
> > >
> > > > The junction tree VAE I mention in section 2.2 would also see the continuous "goodness function" in its BayesOpt routine and so should also obtain information from the bad examples
> > >
> > > If our understanding of Jin et al.'s JT-VAE is correct, the VAE is trained on a fixed 250k molecule dataset (a random subset of ZINC), thus the embeddings that it learns may only be able capture the "positive" parts of molecule space (ZINC itself is not random, it is a "database of commercially-available compounds"). It is true that the BayesOpt part will capture the information of good & bad examples, but it is limited by the learned embedding space which likely only covers the regions around the "positive" (ZINC) parts of molecule space. By contrast in our setup, generation is not limited by this embedding space and has access to the entire molecule space spanned by our block vocabulary.
> > >
> > > We hope this clears up any confusion.

---

> > > > ### Comment · Reviewer_1AuD · 2021-08-16
> > > > **Thanks!**
> > > >
> > > > > We hope this clears up any confusion.
> > > >
> > > > Yes -- many thanks for the speedy reply!

---

> ### Comment · Reviewer_1AuD · 2021-09-03
> **Update at end of the discussion period**
>
> Having read the other reviews and comments, I'm inclined to keep my original score and stand by my original review. I believe this paper proposes interesting ideas and strong results, although the clarity could be improved as mentioned by other reviewers too (e.g. Reviewer UVLM) -- but this should not be hard to fix. The authors also state that they will add further experiments to the paper (e.g. a comparison with a molecular autoencoder approach), which should improve the paper further.
>
> One of the main oddities brought up in the rebuttal was the use of a non-random dataset for the "random" molecule dataset described in Section 4.2.  However, with further clarification, I now realize that this is only used for scoring proxy, and the authors say that they will fix the description of this in the paper, meaning that I now consider this concern resolved.

---

### Official Review · Reviewer_asFX · 2021-07-20

**Rating:** 7
**Confidence:** 3

**Summary:**

This paper propose GFlowNet, a generative method that can turn a given positive reward into a generative policy that samples with a probability proportional to the return.


**Limitations And Societal Impact:**

yes

**Main Review:**

The most interesting part is that this novel generative method is based on flow network and local flow-conditions. And then the authors cast the flow consistency equations into an objective just like casting the Bellman equations into TD objectives. Compared with other methods, GFlowNet is designed especially for diverse candidate generation.

Overall, I vote for accepting. This current submission is overall well written, clearly illustrated and appropriately structured. The motivations are well explained and experimental results are strong.


Question:

* Looking at Figure 4, it seems that MARS has the trend to have a higher average reward if the generative models are trained more than 10^6 molecules. "For GFLowNet and for MARS, the more molecules are visited, the better they become, with slow converge towards the proxy's max reward", which method converges slower? Are there any theoretical/experimental results?

* In equation (9), the hyperparameter $\epsilon$ is added to avoid numerical issues. It would be better if the authors can provide more details on why $\epsilon$  can trade off how much pressure on large versus small flows. And add a brief description about how to setting this hyperparameter for different cases.

UPDATE:
I appreciate the authors' response to my questions and concerns.  I will keep my original score.

**Time Spent Reviewing:**

2.5 hours

---

> ### Author Response · Authors · 2021-08-10
> **Answers to questions**
>
> Thank you for your review, we will incorporate these comments to our paper.
>
> > Looking at Figure 4, it seems that MARS has the trend to have higher average reward if the generative models are trained more than 10^6 molecules. "For GFLowNet and for MARS, the more molecules are visited, the better they become, with slow converge towards the proxy's max reward", which method converge slower? Is there any theoretical/experimental results?
>
> Figure 4 suggests that GFlowNet and MARS both appear to be _able_ to converge in terms of the top molecules found, whereas PPO seems to plateau. However, the top-$k$ reward curves across different values of $k$ point to a larger number of high-reward molecules found by GFlowNet. If we were to train for 10x longer (i.e. to $10^7$ mols), just extrapolating from the curve, it seems unlikely that MARS would surpass GFlowNet. At best, MARS would match GFlowNet in terms of the top-$k$ with smaller $k$, but due to the low diversity of MARS (see Fig 13,14) it seems more likely that MARS would not be able to find as many unique top-k molecules.
>
>
> > In equation (9), the hyperparameter \epsilon is added to avoid numerical issues. It would be better if the authors can provide more details why  can trade off how much pressure on large versus small flows. And add a brief description about how to setting this hyperparameter for different cases.
>
> It is true that this hyperparameter is worth discussing more. We will refine our explanation.
> Ideally, $\epsilon$ should be set to be about as low as the minimum reward one cares about, or close to the minimum useful reward of the environment. This would help the model spend less capacity on the states with flow lower than that.

---

### Decision · Program_Chairs · 2021-09-27

**Decision:**

Accept (Poster)

**Comment:**

After an extensive back-and-forth discussion, all reviewers felt positively about the paper and its contribution, and lean towards acceptance.

However, there were issues regarding clarity in the submitted draft — this has been largely addressed through clarifications from the authors, but it is essential that the discussion here is incorporated into the final version of the manuscript (and in particular, the responses to reviewer 1AuD).